# Transient juvenile demyelination impairs maturation and function of parvalbumin-positive interneurons in the prefrontal cortex

Sara Hijazi[1,2]*, Maria Pascual-García[1], Yara Nabawi[1], Steven A. Kushner[1,3]*

**1** Department of Psychiatry, Erasmus MC, Rotterdam, The Netherlands, **2** Department of Pharmacology, University of Oxford, Oxford, United Kingdom, **3** Department of Psychiatry, Columbia University, New York, New York, United States of America

* sara.hijazi@pharm.ox.ac.uk (SH); sk2602@cumc.columbia.edu (SAK)

## Abstract

Recent studies have highlighted axonal myelination as a common feature of parvalbumin-positive (PV) interneurons throughout the cerebral cortex. However, the precise function of PV interneuron myelination remains incompletely understood. In this study, we used the cuprizone model of demyelination in mice to investigate how PV interneuron myelination might influence their neuronal physiology. Specifically, we examined whether impairing myelination from postnatal day 21 onwards, during a critical neurodevelopmental period of the prefrontal cortex (PFC), can affect PV inter-neuron maturation and function. Using whole-cell patch-clamp recordings to examine intrinsic properties of PV interneurons in the PFC, we found that juvenile demye-lination in mice induced robust alterations of PV interneuron firing patterns. Spe-cifically, we observed that demyelination caused an impairment in the ability of PV interneurons to sustain high-frequency firing associated with a substantial decrease in Kv3-specific currents. We also found a significant impairment in PV interneuron autaptic self-inhibitory transmission, a feature implicated in temporal control of PV interneuron firing during cortical network activity. Following a remyelination period of 5 weeks, PV interneuron properties were only partially recovered, suggesting that transient juvenile demyelination leads to long-lasting impairments of PV interneuron function. In contrast, adult demyelination had no significant effects on PV interneuron firing properties. Together, our data uncovers a critical period for juvenile myelination as an important factor in PFC PV interneuron development and brain maturation.

## Introduction

Cortical parvalbumin-positive (PV) interneurons, also known as fast-spiking inter-neurons, undergo accelerated developmental maturation, including narrowing of the action potential (AP) waveform and high-frequency firing, during a stereotyped postnatal time window extending between postnatal day (P)14 and P28, which

**Data availability statement:** All relevant data is available within the manuscript and Supporting information files.

**Funding:** This work was supported by the Nederlandse Organisatie voor Wetenschappelijk Onderzoek (NWO) (S.A.K. 013.18.002, https://www.nwo.nl/en/projects/01318002), the ERA-Net for Research Programmes on Rare Diseases (S.A.K. JTC2018-024, https://www.neuron-eranet.eu/projects/OPCphrenia/), the Blaschko Fund (S.H., https://www.pharm.ox.ac.uk/), and the UK Research and Innovation (S.H. EP/Z001358/1, https://gtr.ukri.org/projects?ref=EP%2FZ001358%2F1). The funders had no role in study design, data collection and analysis, decision to publish, or preparation of the manuscript.

**Competing interests:** The authors have declared that no competing interests exist.

**Abbreviations:** ACSF, artificial cerebrospinal fluid; AHP, after-hyperpolarization; AP, action potential; MBP, myelin basic protein; PBS, phosphate-buffered saline; PFA, paraformaldehyde; PFC, prefrontal cortex; PPR, paired-pulse ratio; PV, parvalbumin-positive; RMP, resting membrane potential; TEA, tetraethylammonium.

depends on the developmentally regulated expression of a specific complement of voltage-gated ion channels [1–3]. Another important developmental aspect of PV interneurons is their extensive myelination which emerges during a closely overlapping time window [3–5]. While the role of myelination has been widely studied in long-range projection neurons, the impact of myelination on local PV cortical interneurons is still poorly understood. Recent studies have endorsed the classical view that myelination of PV interneurons functions to increase axonal conduction velocity in a manner similar to that of projection neurons [6], while also being critical for PV interneuron-mediated feedforward inhibition in cortical sensory circuits [7] and modulation of local circuit synchronization [8]. However, the specific role of PV interneuron myelination over the time course of neurodevelopment has yet to be elucidated.

Aberrant PV interneuron maturation has been suggested as a contributor to the pathogenesis of multiple neurodevelopmental diseases, and impairments in the development of prefrontal cortex (PFC) circuits has been at the center of many studies investigating neurodevelopmental disorders in rodents [9–13]. Hence, the maturation of PV interneurons in PFC is of particular significance. Notably, PFC development persists long after adolescence in both humans and rodents [12,14] and PFC-dependent functions, such as cognitive flexibility and emotional memory encoding, start emerging later in development, as has been demonstrated in many rodent studies [12,14,15]. This delayed period of maturation of PFC circuits underlies an important time window necessary for the establishment of specific neuronal circuits within the PFC [9,14,16]. Conversely, it also reveals a transient period during which the PFC is vulnerable to various types of stressors and insults [17].

Here, we investigated how the neurodevelopmental timing of myelination influences PV interneuron maturation in the PFC of mice. Our data reveal that myelination of PV interneurons during a critical period of development is instrumental for PV interneuron maturation and their unique ability to fire at high frequencies. We find that impaired myelination in early adolescence can have long-lasting effects on PV interneuron morphology and function in adulthood in mice.

## Results

### Myelination-dependent critical period for PV interneuron maturation

**Juvenile demyelination affects PV interneuron properties, maturation, and morphology in PFC.** In order to investigate the relationship between myelination and PV interneuron maturation in the PFC, mice were fed a 0.2% cuprizone diet for 6 weeks starting at P21. Mice were then sacrificed and brain slices prepared for whole-cell patch-clamp recording and filling of PV interneurons (Fig 1A). Juvenile cuprizone treatment led to a clear decrease in myelination in adulthood (Fig 1B–1D) and a complete demyelination of PV interneuron axons in the PFC while the density of PV interneurons was unaffected (Fig 1E). Next, we fully reconstructed biocytin-filled patched cells and quantified their axonal and dendritic processes (Fig 1F–1H). None of the PV interneurons tested from cuprizone-treated mice showed myelinated segments (0 out of 7 cells). We found a significant decrease in total axonal length

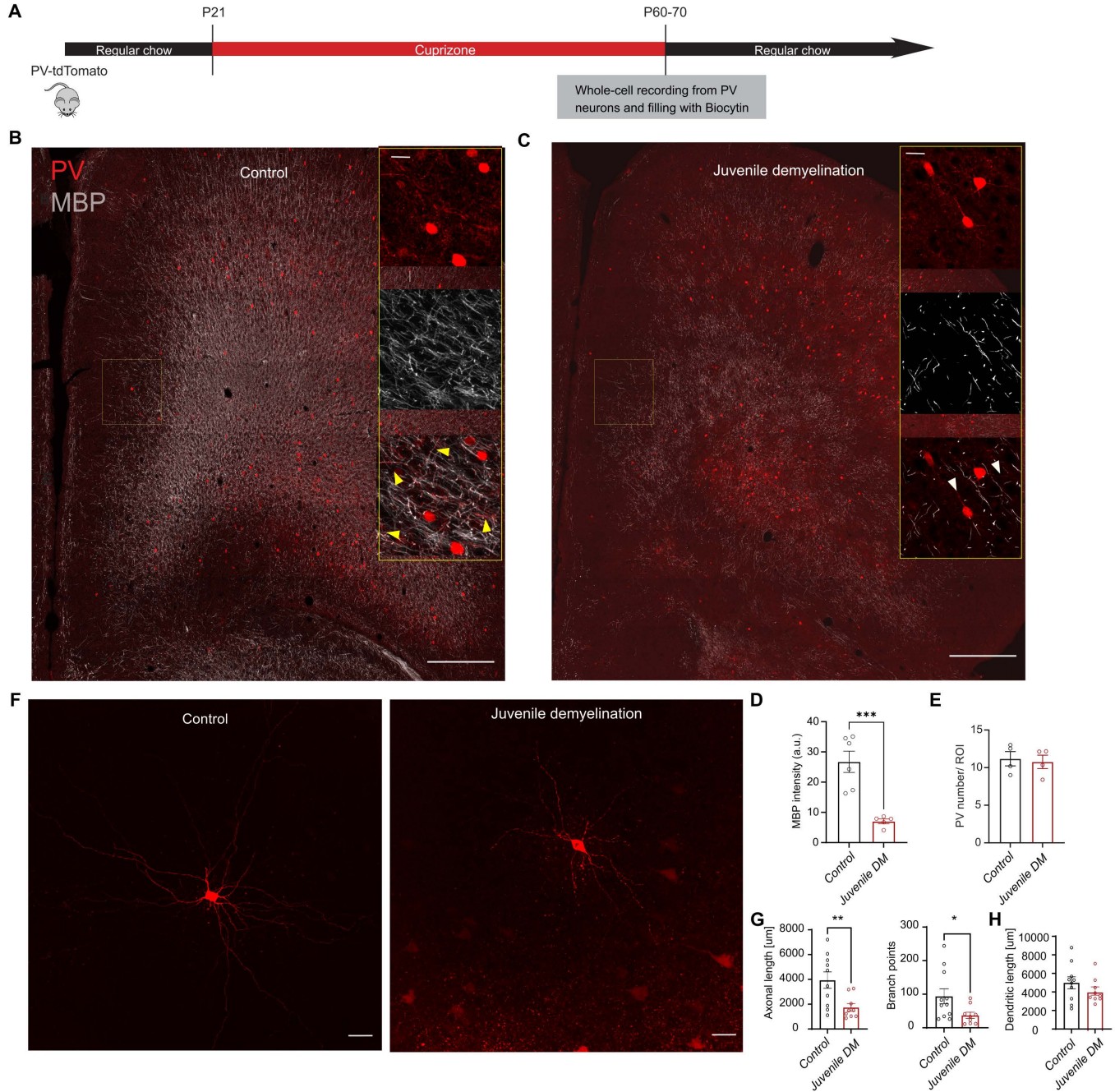

**Fig 1. Juvenile demyelination leads to a decrease in axonal complexity of PFC PV interneurons. A.** Experimental design to induce juvenile demyelination in PV-tdTomato mice **B.** Confocal tiled image of PFC in PV-tdTomato animal (tdTomato+, red) overlaid with myelin basic protein (MBP, gray); scale bar: 300 μm. *Right inset.* 40× image showing myelinated PV interneuron axons in control conditions (MBP, gray). Yellow arrowheads point to myelinated PV interneuron axons; scale bar: 30 μm. **C**. Confocal overview image of the PFC region immunolabeled for MBP expression showing the loss of myelin following 6 weeks of 0.2% cuprizone-diet. *Right inset.* 40× image showing no myelination of PV interneuron axons in cuprizone conditions. Scale bars same as panel B. **D.** Quantification of the overall myelin intensity in PFC of both groups shows a clear loss of myelin in the juvenile demyelination group (*t test; n = 6/5 mice per group, ***p < 0.001*). **E**. Quantification of the total number of PV interneurons in the PFC region shows no difference between control (black) and cuprizone-treated (red) mice (*t test; n = 4 mice per group, p = 0.7577*). The number of PV-positive cells was quantified in high magnification images. **F**. Example image of a biocytin-labeled PFC PV interneuron from a control mouse (left) and from cuprizone-treated mouse (right); scale bar: 50 μm. **G**. *Left.* Total proximal axonal length is decreased in cuprizone-treated PFC PV interneurons compared with control cells (*t test;*

*n = 10/9 cells per group, \*\*p < 0.01). Right.* The number of branch points in PV interneuron axons from cuprizone-treated mice was significantly decreased compared to control mice (*t test; n = 10/9 cells per group, \*p < 0.05*). **H.** Total dendritic lengths were unaltered by juvenile demyelination (*t test; n = 10/9 cells per group, p = 0.2856*). The data displayed in (D), (E), (G), and (H) can be found in S1 Fig.

and axonal branching of PV interneurons from mice that underwent juvenile demyelination (Fig 1F). The total dendritic length of PV interneurons were unchanged (Fig 1G–1I).

Given that demyelination during adolescence led to changes in PV interneuron morphology, we next aimed to characterize whether PV interneuron physiology was correspondingly affected. To that end, we performed current clamp recordings from PFC PV interneurons following juvenile demyelination. Our data revealed alterations in both passive and active membrane properties of PFC PV interneurons in mice that underwent juvenile demyelination, indicative of an immature PV interneuron phenotype (Fig 2). Specifically, there was a significant increase in input resistance and sag amplitude (Fig 2A–2F). The AP waveform was also affected (Fig 2G), showing a substantial increase in AP width, decay time, and after-hyperpolarization (AHP) duration (Fig 2I–2L). Furthermore, there was a significant decrease in the firing frequency of PV interneurons at high-current injections, along with an impairment in the maximum firing frequency (Fig 2M–2O). Strikingly, the difference in firing frequency was apparent from 50 Hz onwards (Fig 2N), in which 38.4% (20 out of 52) of PV interneurons from mice with juvenile demyelination exhibited a failure to sustain repetitive firing, compared to only 11.6% (5 out of 43) of the cells from the control group (Fisher's Exact Test, \*\*p = 0.004; Fig 2P and 2Q). Taken together, these data suggest that juvenile demyelination impairs PV interneuron's ability to fire at high frequencies in the PFC.

**Shiverer PV interneurons have similar properties to juvenile demyelinated PV interneurons.** In order to independently confirm the impact of juvenile demyelination on the maturation of PV interneurons in the PFC, we made use of *shiverer* mice, harboring a germline homozygous deletion of MBP impaired myelination. We performed whole-cell recordings from *shiverer* and wild-type littermate PV interneurons at P60-75 (Fig 3A and 3B). PV interneurons in the PFC of *shiverer* mice exhibited considerable similarity to those of mice that underwent juvenile demyelination (Fig 3C–3N). Specifically, the sag amplitude, AP half-width, AP decay time, and AHP time were all significantly increased compared to WT mice (Fig 3E, 3K, 3L, and 3N). We found no differences in input resistance, resting membrane potential (RMP), or rheobase (Fig 3D and 3G). Finally, when we examined the firing properties of PV interneurons, we observed a clear decrease in firing frequency at increased current injections, along with a decrease in the maximum firing frequency in *shiverer* mice (Fig 3O–3S). This was due to the fact that 37.1% (14 out of 37) of the cells from *shiverer* mice could not sustain high frequency firing compared to 4% (1 out of 25) of their WT littermates (Fisher's Exact Test, \*\*p = 0.004; Fig 3R and 3S).

**Adult demyelination has no significant effect on intrinsic excitability of PV interneurons.** Are the observed firing impairments at high frequencies due to the loss of myelin during a critical period of development or solely caused by the demyelination of PV interneurons axons? To address this, we recorded from PV interneurons from mice that underwent adult demyelination, in which cuprizone treatment was initiated at P60 instead of P21 (Fig 4A–4C). PV interneurons from mice with adult demyelination showed a more hyperpolarized resting membrane potential and decreased rheobase, with no alterations in firing properties (Fig 4D and 4G). Importantly, input resistance and sag amplitude remained unchanged following adult demyelination (Fig 4E and 4F). Furthermore, the AP waveform and firing frequency of PV interneurons was also unaffected by adult demyelination, in contrast to what was observed following juvenile demyelination and in *shiverer* mice (Figs 4I–4N and S1). Finally, the maximum firing frequency of PV interneurons following adult demyelination also remained intact (Fig 4P and 4Q). Specifically, only 10.0% (2 out of 20) of cells from mice with adult demyelination showed a failure to sustain high firing frequency at high current injections, which was similar to control mice (4.0%, 1 out of 25) (Fisher's Exact Test, *p = 0.577*; Fig 4R and 4S). Together, our data indicates that adult demyelination has no

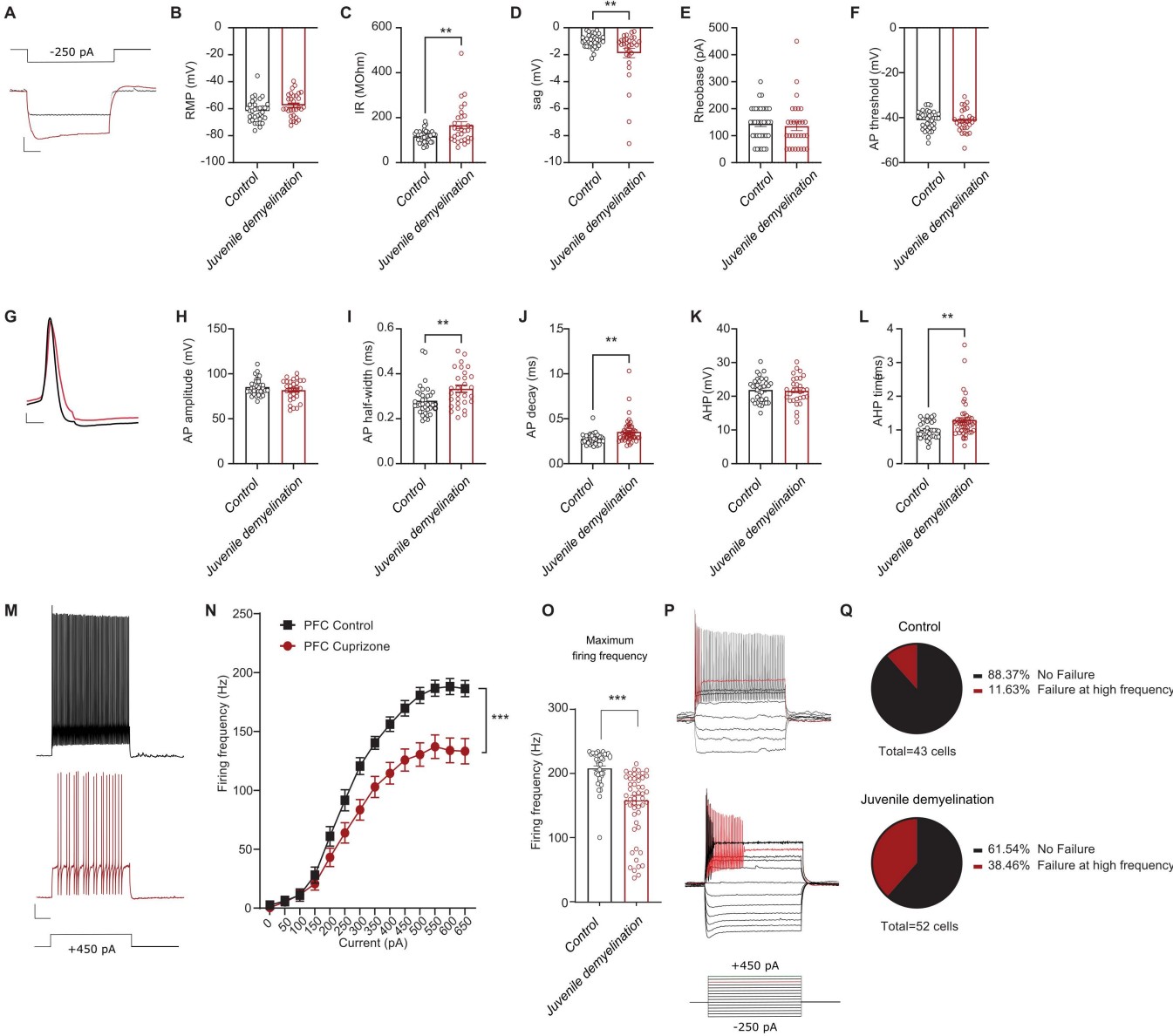

**Fig 2. Juvenile demyelination impairs the maturation of PV interneurons in the PFC. A**. Representative traces of voltage responses following a hyperpolarizing step from control (black) and cuprizone-treated (red) mice illustrating an increased input resistance and sag in mice that underwent juvenile demyelination. Scale bar: 100 ms, 10 mV. **B–F**. Summary data showing the averaged (±s.e.m) of the following intrinsic properties: **(B)** Resting membrane potential (RMP) (*t test; n = 36/30 cells from 9/8 mice per group, p = 0.0511*), **(C)** Input resistance (IR) (*t test; n = 36/30 cells from 9/8 mice per group, **p < 0.01*), **(D)** Sag (*t test; n = 36/30 cells from 9/8 mice per group, **p < 0.01*), **(E)** Rheobase (*t test; n = 36/30 cells from 9/8 mice per group, p = 0.6853*) and **(F)** Action potential (AP) threshold (*t test; n = 36/30 cells from 9/8 mice per group, p = 0.7831*). **G**. A representative trace of a single AP illustrating a wider and slower AP in cuprizone-treated mice (red) compared to control (black). Scale bar: 1 ms, 10 mV. **H–L**. Summary data showing the averaged (± s.e.m) of the following AP waveform properties: **(H)** AP amplitude (*t test; n = 36/30 cells from 9/8 mice per group, p = 0.1534*), **(I)** AP half-width (*t test; n = 36/30 cells from 9/8 mice per group, **p < 0.01*), **(J)** AP decay (*t test; n = 36/30 cells from 9/8 mice per group, **p < 0.01*), **(K)** After-hyperpolarization (AHP) amplitude (*t test; n = 36/30 cells from 9/8 mice per group, p = 0.7903*) and **(L)** AHP time (*t test; n = 36/30 cells from 9/8 mice per group, **p < 0.01*). **M**. Representative traces of voltage responses following +450 pA current injection. Scale bar: 100 ms, 10 mV. **N**. Average action potential (AP) frequency in response to 0-650 pA current steps illustrating a significant decrease in PV interneuron firing frequency in cuprizone-treated mice. (*group x current two-way repeated measures: n = 36/30 cells from 9/8 mice per group: F(13,836) = 4.00, p < 0.001*). **O**. Summary data of the maximum firing frequency per group (*t test; n = 35/29 cells from 9/8 mice per group, ***p < 0.001*). **P**. Example traces of two different PV interneurons that are unable to maintain high-frequency firing at increased current injection. The upper trace shows a cell that can sustain its firing at low current but not at high, whereas the

lower trace shows a cell that cannot sustain its firing at any current step. **Q**. Percentage of cells that failed to maintain high frequency firing at >500 pA in both groups reveals a clear increase in cuprizone-treated mice (*Fisher's exact test; **p = 0.0045*). Data was obtained from voltage recordings with 500 ms current injections ranging from −300 to 650 pA. The data displayed in (B–F), (H–L), and (N–O) can be found in S1 Table.

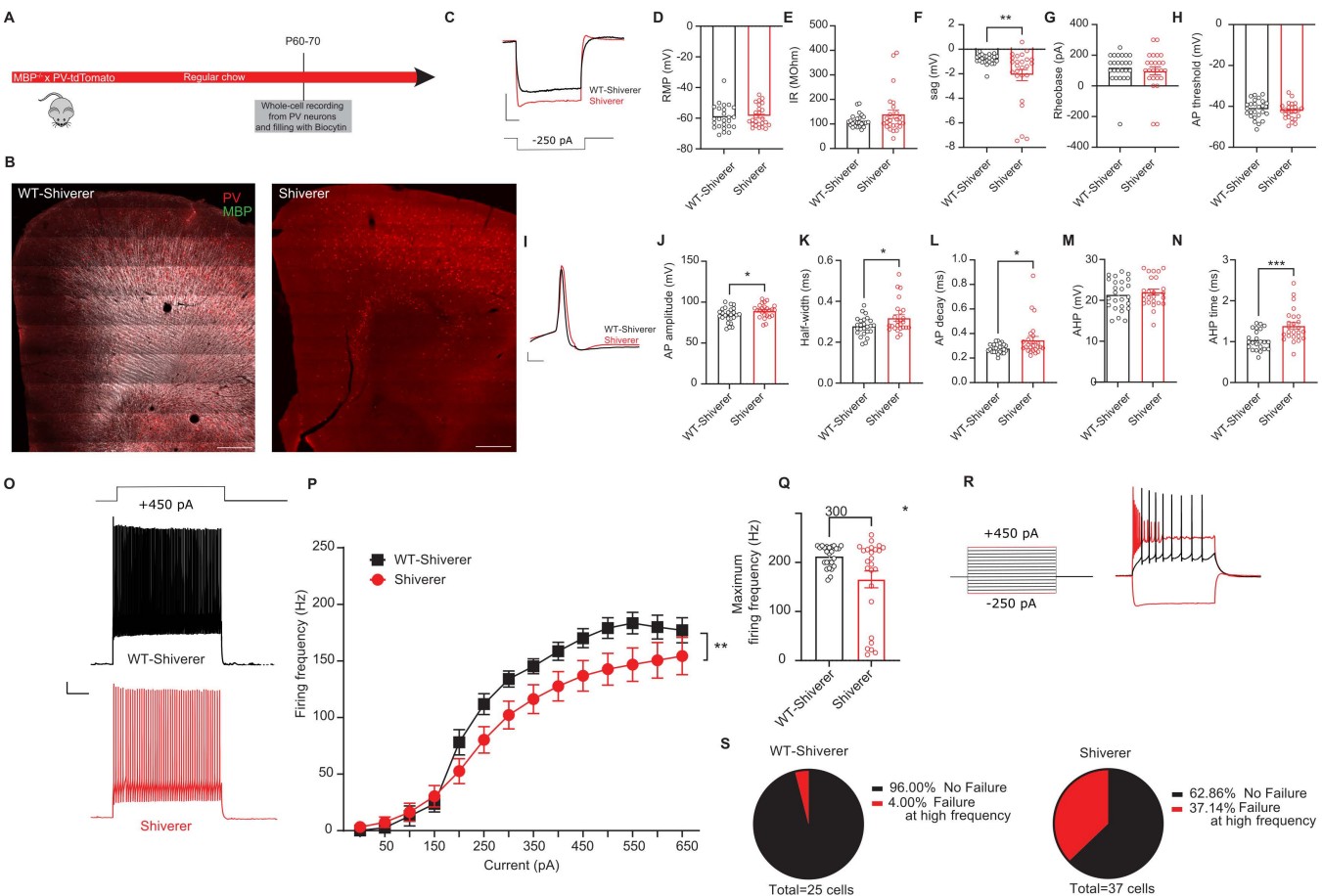

**Fig 3. PV interneurons from Shiverer mice show many similarities with the ones from mice that underwent juvenile demyelination. A.** Experimental design for MBP$^{-/-}$ (Shiverer)-PV-tdTomato mice **B**. *Left.* Confocal overview image of PFC in control PV-tdTomato animal (tdTomato+, red) overlaid with myelin basic protein (MBP, gray). *Right.* Confocal overview image immunolabeled for MBP expression showing the absence of myelination in the PFC of Shiverer mice. Scale bar: 300 μm. **C**. Representative traces of voltage responses following a hyperpolarizing step from control (black) and Shiverer (light red) mice illustrating an increased sag in mice lacking MBP from birth. Scale bar: 100 ms, 10 mV**. D–H**. Summary data showing the averaged (± s.e.m) of the following intrinsic properties: **(D)** Resting membrane potential (RMP) (*t test; n = 25/25 cells from 9 mice per group, p = 0.6752*), **(E)** Input resistance (IR) (*t test; n = 25/25 cells from 9 mice per group, p = 0.1407*), **(F)** Sag (*t test; n = 25/25 cells from 9 mice per group, **p = 0.0072*), **(G)** Rheobase (*t test; n = 25/25 cells from 9 mice per group, p = 0.5406*) and **(H)** Action potential (AP) threshold (*t test; n = 25/25 cells from 9 mice per group, p = 0.4461*). **I**. A representative trace of a single AP illustrating a wider and slower AP in Shiverer mice (light red) compared to control (black). Scale bar: 1 ms, 10 mV. **J–N**. Summary data showing the averaged (± s.e.m) of the following AP waveform properties: **(J)** AP amplitude (*t test; n = 25/25 cells from 9 mice per group, *p = 0.0405*), **(K)** AP half-width (*t test; n = 25/25 cells from 9 mice per group, *p = 0.0278*), **(L)** AP decay (*t test; n = 25/25 cells from 9 mice per group, *p = 0.0258*), **(M)** After-hyperpolarization (AHP) amplitude (*t test; n = 25/25 cells from 9 mice per group, p = 0.5188*) and **(N)** AHP time (*t test; n = 25/25 cells from 9 mice per group, ***p = 0.0007*). **O**. Representative traces of voltage responses following +450 pA current injection. Scale bar: 100 ms, 10 mV. **P**. Average action potential (AP) frequency in response to 0–650 pA current steps illustrating a significant decrease in PV interneuron firing frequency in Shiverer mice (*group x current two-way repeated measures: n = 25/25 cells from 9 mice per group: F(13,624) = 2.303, **p < 0.0056*). **Q**. Summary data of the maximum firing frequency per group (*t test; n = 25/25 cells from 9 mice per group, *p = 0.0109*). **R**. Example trace of a PV interneuron that is unable to maintain high-frequency firing at increased current injection. **S**. Percentage of cells that failed to maintain high-frequency firing at > 500 pA in both groups reveals a clear increase in Shiverer mice (*Fisher's exact test; **p = 0.0041*). The data displayed in **(D-H)**, (J-N) and (P-Q) can be found in S2 Table.

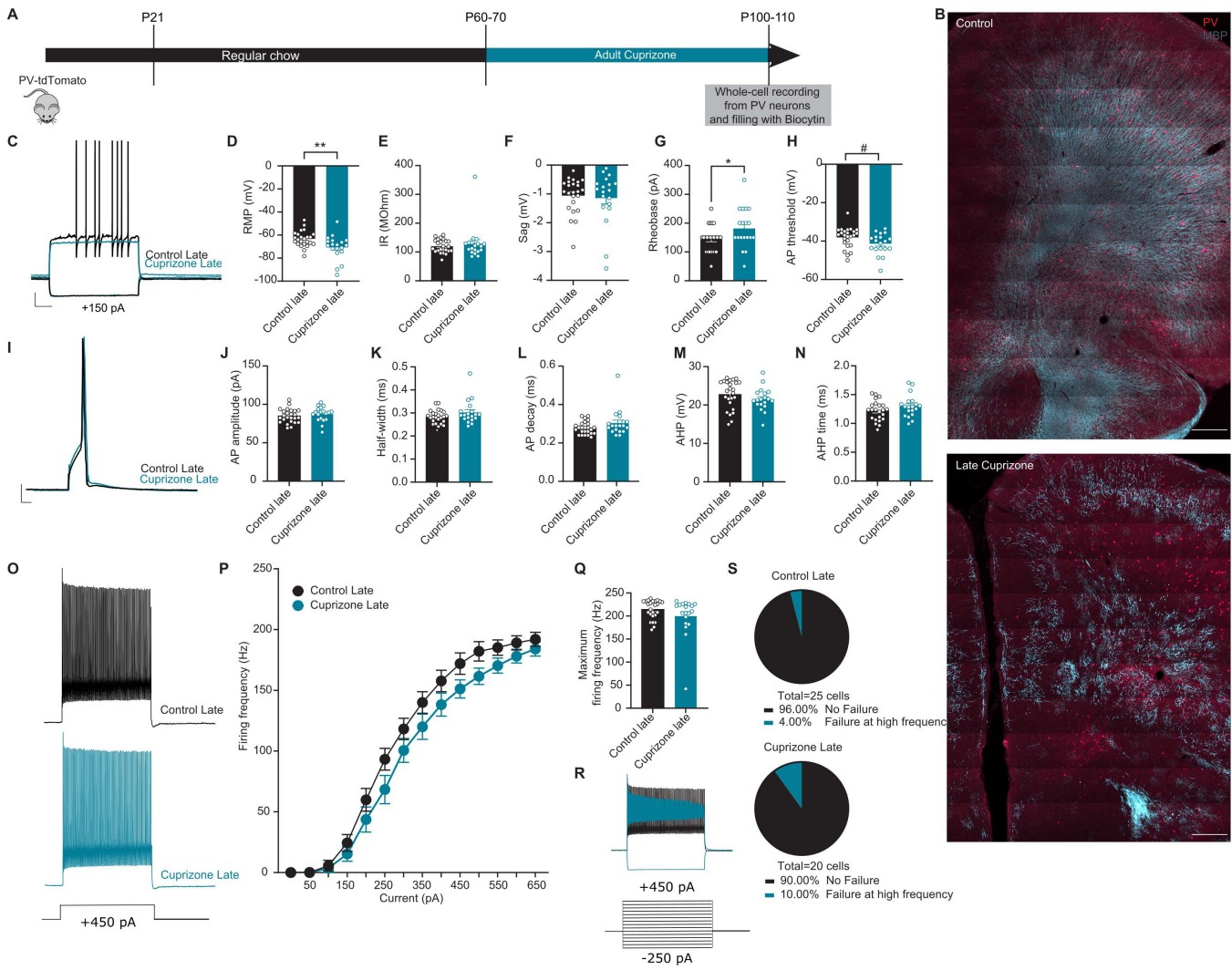

**Fig 4. Adult demyelination has no effect on PV interneuron's firing rate or failures in autaptic responses. A.** Experimental design for adult demyelination in PV-tdTomato mice. **B**. *Upper panel*. Confocal overview image of PFC in PV-tdTomato animal (tdTomato+, red) overlaid with myelin basic protein (MBP, gray). *Lower panel.* Confocal overview image immunolabeled for MBP expression showing the effect of adult demyelination on the PFC region following 6 weeks of 0.2% cuprizone diet. Scale bar: 300 μm. **C**. Representative traces of voltage responses following a depolarizing step from control (Control Late, black) and adult demyelination (Cuprizone Late, blue) mice illustrating an increase in Rheobase in mice following late cuprizone treatment. Scale bar: 100 ms, 10 mV. **D–H**. Summary data showing the averaged (±s.e.m) of the following intrinsic properties: **(D)** Resting membrane potential (RMP) (*t test; n = 24,19 cells from 8–6 mice per group, **p = 0.0086*), **(E)** Input resistance (IR) (*t test; n = 24,19 cells from 8–6 mice per group, p = 0.5066*), **(F)** Sag (*t test; n = 24,19 cells from 8–6 mice per group, p = 0.6607*), **(G)** Rheobase (*t test; n = 24,19 cells from 8–6 mice per group, *p = 0.0336*) and **(H)** Action potential (AP) threshold (*t test; n = 24,19 cells from 8–6 mice per group, #p = 0.0541*). **I**. A representative trace of a single AP illustrating no difference in AP waveform between late cuprizone (blue) compared to control (black). Scale bar: 1 ms, 10 mV. **J–N**. Summary data showing the averaged (± s.e.m) of the following AP waveform properties: **(J)** AP amplitude (*t test; n = 24,19 cells from 8–6 mice per group, p = 0.3837*), **(K)** AP half-width (*t test; n = 24,19 cells from 8–6 mice per group, p = 0.1139*), **(L)** AP decay (*t test; n = 24,19 cells from 8–6 mice per group, p = 0.1004*), **(M)** After-hyperpolarization (AHP) amplitude (*t test; n = 24,19 cells from 8–6 mice per group, p = 0.2520*) and **(N)** AHP time (*t test; n = 24,19 cells from 8 to 6 mice per group, p = 0.1543*). **O**. Representative traces of voltage responses following +450 pA current injection. **P**. Average action potential (AP) frequency in response to 0–650 pA current steps showing no difference in PV interneuron firing frequency in late cuprizone (blue) mice compared to control (black) mice. (*group x current two-way repeated measures: n = 24/19 cells from 8–6 mice per group: F(13,455) = 1.396, p = 0.1569*). **Q**. Summary data of the maximum firing frequency per group (*t test; n = 24,19 cells from 8–6 mice per group, p = 0.1581*). **R**. Example trace of a PV interneuron that is unable to maintain high-frequency firing at increased current injection. **S**. Percentage of cells that failed to maintain high frequency firing at >500 pA was very low after adult demyelination (*Fisher's exact test; p = 0.5772*). The data displayed in **(D–H)**, (J–N), and (P–Q) can be found in S4 Table.

effect on the AP waveform and the sustained firing properties at high frequencies of PV interneurons, further validating that the aberrant firing observed in the juvenile demyelination group is due to the loss of myelin at a critical age of PFC development.

### What are the mechanisms underlying impaired high-frequency firing of PV interneurons following juvenile demyelination?

**PV interneurons show a significant decrease in Kv3 currents and Kv3 expression following juvenile demyelination.** Next, we sought to unravel the mechanism by which juvenile demyelination could be impairing PV interneurons firing at high frequency. The ability of PV interneurons to sustain high-frequency firing results from their very fast AP properties endowed by a prominent expression of Kv3 channels [18]. Previous studies have identified an upregulation of K+ channel subunits of the Kv3 subfamily during the second and third postnatal week that drives these specific features of fast-spiking PV interneurons [1,19]. Therefore, we aimed to examine whether the immature phenotype displayed by PV interneurons following juvenile demyelination might be explained by changes in K+ currents in these cells. To that end, we performed voltage-clamp recordings from PFC PV interneurons following juvenile demyelination (Fig 5). The amplitude of the K+ currents from +10 mV to +60 mV was significantly lower in PV interneurons following juvenile demyelination (Fig 5A and 5B). To verify that these changes in K+ currents were at least partly due to changes in Kv3 channel activity, we performed the same voltage-clamp recordings in the presence of tetraethylammonium (TEA) (1 mM), which results in a relatively selective block of Kv3 potassium channel [20] (Fig 5C). The data confirmed a decrease in TEA-sensitive currents in the cuprizone group, with a significant two-way interaction of genotype and voltage step (Fig 5D). Next, we used immunolabeling against Kv3.1b channels to quantify Kv3 channel expression in PV interneurons from the cuprizone group (Fig 5E). We observed a significant reduction of Kv3 immunofluorescence along PV interneurons of the PFC in mice that underwent juvenile demyelination compared to controls, while PV expression and cell density were unaltered (Figs 5F–5J, 1H, and 1I). Together, our results uncover that juvenile demyelination leads to a decrease in Kv3 channel expression and activity within PV interneurons of the PFC.

**Positive allosteric modulation of Kv3 conductance results in a partial rescue of PV interneuron properties following juvenile demyelination.** In order to test whether decreased Kv3 activity contributes to the observed phenotypes reported in PV interneurons after juvenile demyelination, we used AUT00201, a positive allosteric modulator of Kv3 potassium currents [21], to evaluate whether PV interneuron properties are causally related to the observed reduction of Kv3 channel expression following juvenile demyelination. To that end, we measured intrinsic properties of PV interneurons at baseline and following 5 min of bath application of AUT00201 (1 μM) (Fig 6). We found that AUT00201 decreased the rheobase and rescued AP half-width of PV interneurons from mice with juvenile demyelination (Fig 6A and 6B). Intriguingly, AUT00201 decreased the firing threshold of PV interneurons in both groups, while the remaining properties of PV interneurons were unaltered (S2 Fig).

As AUT00201 rescued AP half-width in mice with juvenile demyelination, we examined whether the I–V relationship of PV interneurons could be rescued by bath application of AUT00201 (1 μM). However, when we looked at the firing frequency curves, we did not see any overall significant improvement (S2F Fig). After a close examination of the effect on AUT00201 on individual cell firing patterns, we found that it increased firing in 61.5% (8 out of 13) of PV interneurons from juvenile demyelination mice while having no effect in 23.0% of cells (3 out of 13) and decreasing the firing in 15.4% of the cells (2 out of 13) (Figs 6C and S3). Conversely, in control cells, AUT00201 (1 μM) increased PV interneuron firing in only 8.3% of the cells (1 out of 12), whereas it had no effect on the firing properties in 58.3% of the cells (7 out of 12). AUT00201 even induced an impairment in firing at high current injections in 33.3% of cells from control mice (4 out of 12) (Fisher's Exact Test, *p = 0.024; Fig 6C and 6D). Analysis of only the positively affected cells in the juvenile demyelination group revealed a significant increase in the firing frequency of PV interneurons after AUT00201 application (Fig 6E). Interestingly, bath application of the Kv3 modulator AUT00201 (1 μM) had no effect on PV properties in the adult demyelination

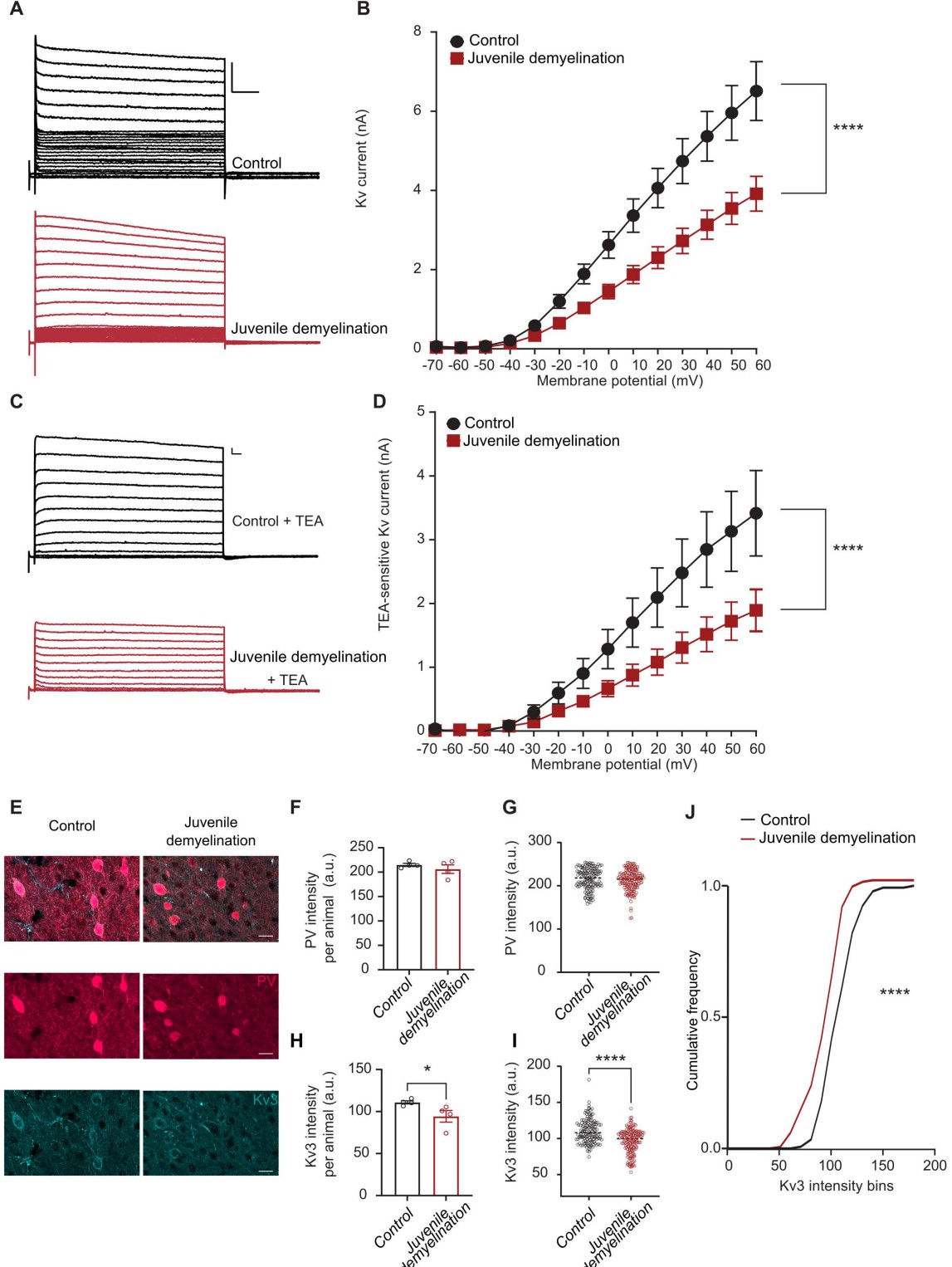

**Fig 5. Juvenile demyelination leads to a substantial decrease in potassium currents in PV interneurons of the PFC. A.** Representative traces of potassium currents evoked with 10 mV potential steps from −70mV to +60 mV in PFC PV interneurons from control (black) and cuprizone (red) mice. Scale: 500 pA, 100 ms. **B.** I–V curves showing a significant decrease in Kv amplitude in PV interneurons from mice that underwent juvenile

demyelination. (*group x voltage two-way repeated measures: n = 12,18 cells from 5 mice per group: F(13,364) = 10.45, p < 0.0001*). **C**. Representative traces of TEA-sensitive potassium currents (1mM). **D**. TEA-sensitive currents were specifically decreased in PV interneurons from mice that received cuprizone treatment compared to control mice (*group x voltage two-way repeated measures: n = 10,14 cells from 5 mice per group: F(13,286) = 5.082, p < 0.0001*). **E**. Representative images of double immunostaining of PV interneurons (red) and Kv3.1b (cyan) in the PFC in control and cuprizone mice; scale bar: 50 μm. **F, G**. PV intensity was not altered following juvenile demyelination (*t test of (F); n = 4 mice per group, p = 0.3888; t test of (G); n = 145/170 cells from 4 mice per group, p = 0.3067*). **H–J**. Kv3.1b intensity was significantly decreased in PV interneurons from cuprizone mice (red) compared to control mice (black) (*t test of (H); n = 4 mice per group, p = 0.0286; t test of (I); n = 145/170 cells from 4 mice per group, p < 0.0001, Kolmogorov–Smirnov test of (J); p < 0.0001*). The data displayed in (B), (D) (F–G), and (H–I) can be found in S5 Table.

group (S4 Fig). Taken together, these data imply that a change in Kv3 activity might account for some of the impairment observed after juvenile demyelination in PV interneurons inability to fire at high frequencies.

**Juvenile demyelination induces a loss of functional autapses and impairs autaptic plasticity.** Recent studies have supported a role for autapses, synaptic inputs from individual PV interneurons onto themselves, in regulating their fast-spiking properties and modulation of network oscillations [22–24]. Accordingly, in the next set of experiments, we set out to explore whether juvenile demyelination altered autaptic transmission, possibly underlying the observed impairments in sustained repetitive firing. To that end, we recorded inhibitory autaptic postsynaptic responses in PFC PV interneurons at P60–75 (Fig 7A–7D). Significantly fewer PV interneurons from mice with juvenile demyelination exhibited an autaptic response (53.8%; 28 out of 52) compared to the control group (75.9%; 44 out of 58) (Fisher's Exact Test, *p = 0.017; Fig 7B). However, among PV interneurons with an autaptic response, autaptic transmission appeared normal (Fig 7C and 7D).

Next, we went to back to the reconstructed cells and examined whether they had morphological evidence of autapses, defined as colocalization between PV interneuron axon terminals and its own soma or proximal dendrite (Fig 7E and 7F; S1 Video). We observed autaptic morphologies in 88.2% (15 out of 17) of control mice, but only 70.8% (17 out of 24) of PV interneurons from mice with juvenile demyelination, suggesting that an important mechanism underlying the loss of autaptic transmission following juvenile demyelination may involve overt structural loss of autapses (Fisher's Exact Test, **p = 0.007; Fig 7F). In PV interneurons without an autaptic morphology, we never observed a functional autaptic response (n = 9 cells). Furthermore, in mice with juvenile demyelination, even among PV interneurons with morphological evidence of autapses, over one-quarter showed no functional autaptic response (6 out of 24 cells in juvenile demyelination group compared to 0 out of 17 control cells, Fisher's Exact Test for all variables, ****p < 0.0001; Fig 7E and 7F). We also recorded autaptic responses from PV interneurons from *shiverer* mice. 59.5% (15 out of 37) of PV interneurons from *shiverer* mice showed no autaptic responses compared to only 16.0% (4 out of 25) in their wild-type littermates (Fisher's Exact Test, p = 0.051; Fig 7G and 7H). Amplitude and decay of the autaptic responses were also not affected (Fig 7I and 7J). Finally, to confirm that our results reflect an impairment due to the loss of myelin during development, we recorded autaptic responses from PV neurons following adult demyelination and we found no difference in the percentage of cells showing an autaptic response in mice from the adult demyelination group (65.0%; 7 out of 20) compared to control mice (72.0%; 7 out of 25) (*Fisher's Exact Test, p = 0.749*; Fig 7K–7N).

In view of the observed impairment of sustained high-frequency firing following juvenile demyelination (Fig 2), we next examined whether autaptic plasticity at high frequency is also affected. Evoked autaptic responses were recorded following 100 APs at 200 Hz in voltage-clamp mode. Although control mice showed clear paired-pulse facilitation of autaptic transmission (IPSC2/1; mean paired-pulse ratio (PPR): 1.23), mice with juvenile demyelination showed no evidence of plasticity with a mean PPR of 1.02 (Fig 8A and 8B). This impairment in autaptic plasticity was also observed in *shiverer* mice (Fig 8C) but not in the adult demyelination group (Fig 8D). Furthermore, we noted a significant increase in failure rate at 200 Hz of the subsequent evoked autaptic responses in the juvenile demyelination group where 61.1% (11 out of 18) of PV interneurons showed at least 1 failed response compared to only 13.3% (2 out of 15) in the control group (Fisher's Exact Test, *p = 0.011; Fig 8E and 8F). In *shiverer* mice 26.7% (4 out of 15) of PV interneurons showed at least one failed

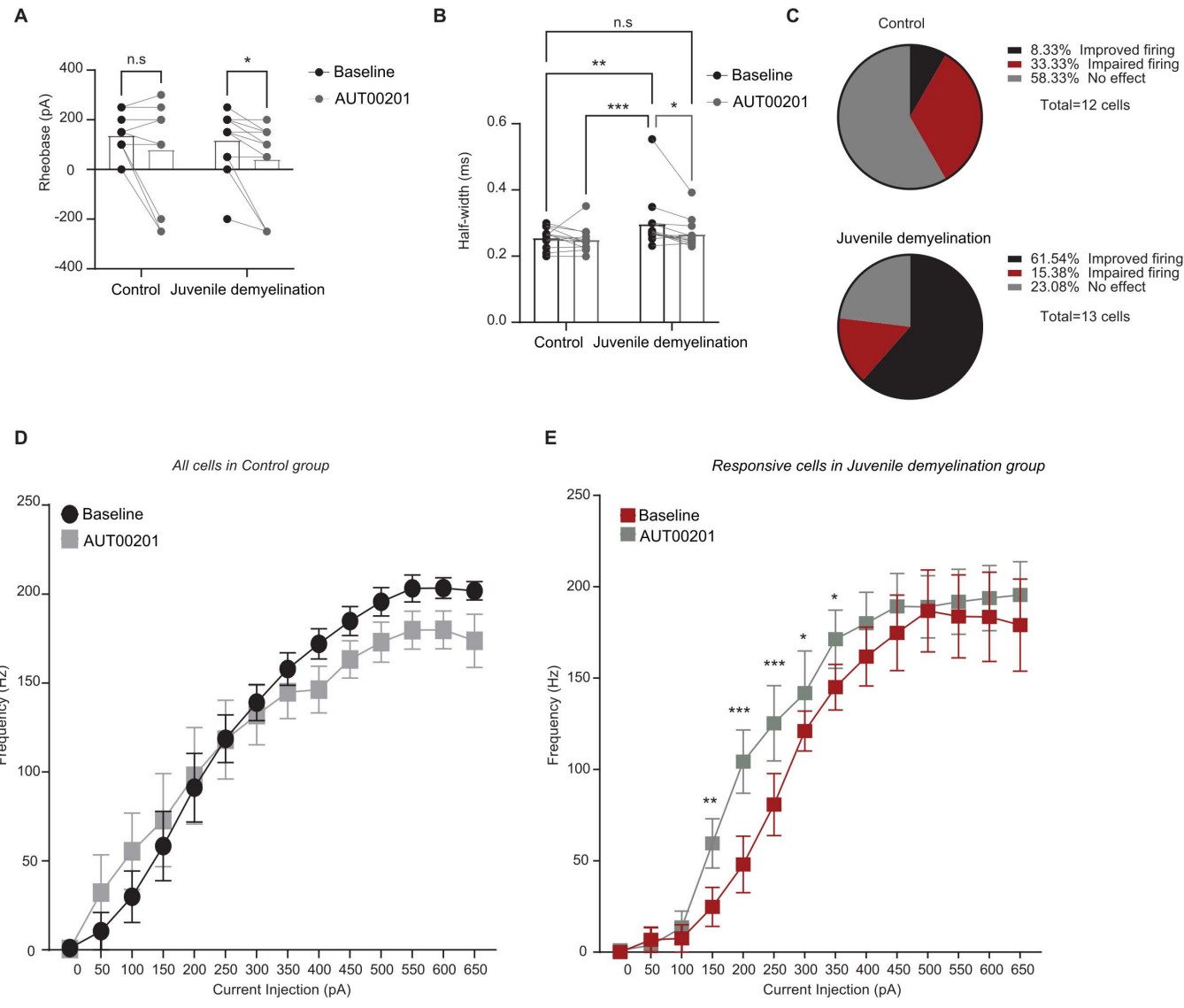

**Fig 6. AUT00201 (1 μM) can rescue AP width in cuprizone mice. A**. Bath application of AUT00201 1 μm decreased the rheobase of PV interneurons from cuprizone mice but not control mice (*group x AUT00201 two-way repeated measures: n = 12,13 cells from 4 mice per group: F(1,12) = 0.1816, p = 0.6775; AUT00201 effect: F(1,12) = 7.122, \*p = 0.0205. Post-hoc LSD test: \*p < 0.05, n.s. p = 0.0957*). **B**. AUT00201 restores AP half-width in PV interneurons from cuprizone mice, while having no effect on cells from control mice (*group x AUT00201 two-way repeated measures: n = 12,13 cells from 4 mice per group: F(1,12) = 2.774, p = 0.1217; AUT00201 effect: F(1,12) = 5.685, \*p = 0.0345. Post-hoc LSD test: \*p < 0.05, \*\*p < 0.01, \*\*\*p < 0.001, n.s. p = 0.3148*). **C**. Pie charts reflecting the effect of AUT00201 on PV interneurons firing frequency in control (upper) and cuprizone (lower panel) mice. **D**. Average action potential (AP) frequency in response to 0–650 pA current steps suggesting an impairment at high frequencies in control cells after bath application of AUT00201 (1 μM) (gray) (*group x current two-way repeated measures: n = 13 cells from 4 mice: p = 0.1369*). **E**. In the responsive cells from cuprizone mice (red) a significant increase in PV interneuron firing frequency at lower current steps (gray) was observed (*group x current two-way repeated measures: n = 8 cells from 4 mice: p = 0.05287; AUT00201 effect: F(1,60) = 5.050, \*p = 0.0457. Post-hoc LSD test: \*p < 0.05, \*\*p < 0.01, \*\*\*p < 0.001*). The data displayed in (A), (B) (D), and (E) can be found in S6 Table.

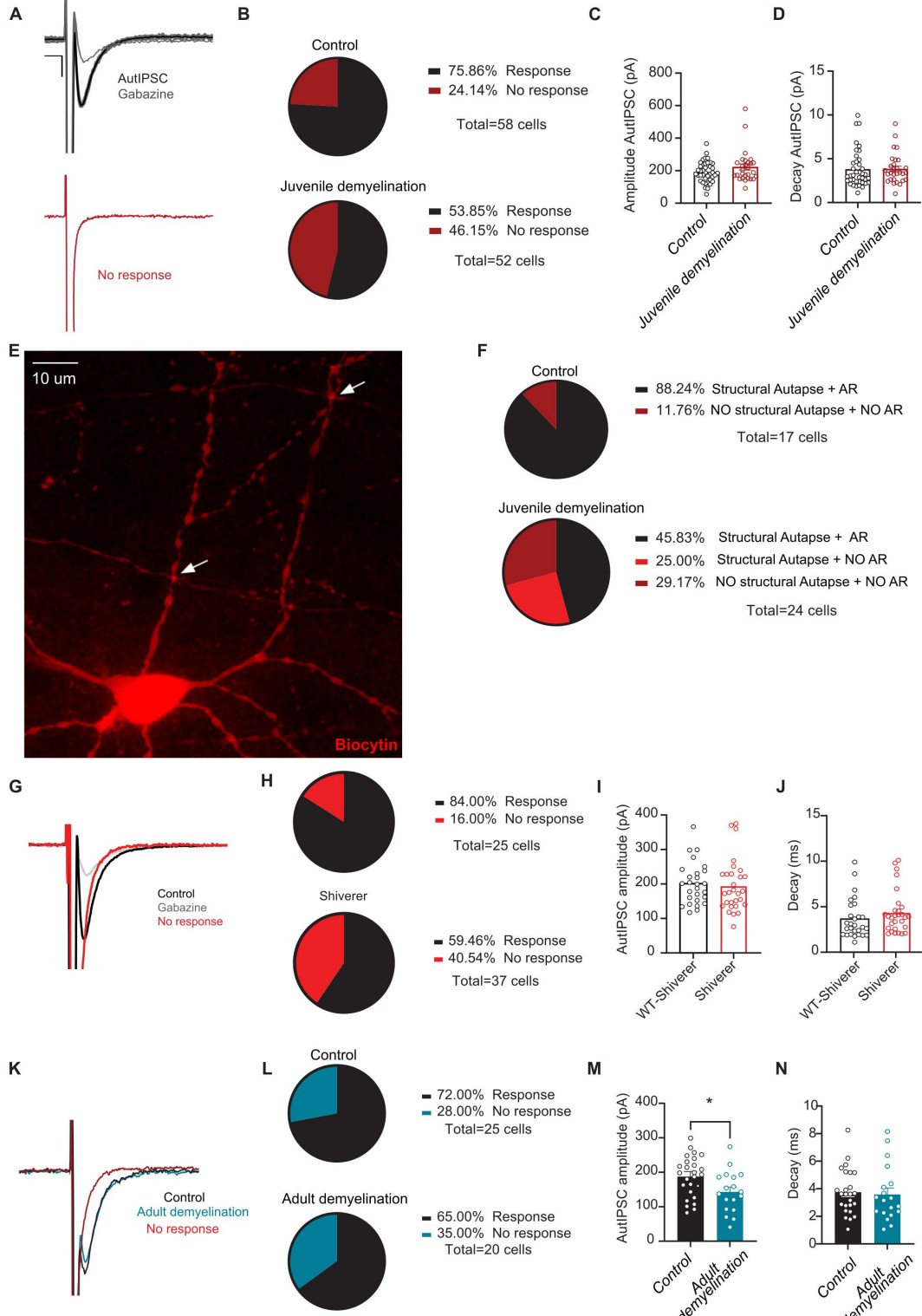

**Fig 7. Juvenile demyelination induces a loss of functional autapses. A**. *Upper panel*. Example traces (average of 8 sweeps) of unitary autaptic inhibitory postsynaptic currents triggered by voltage steps from holding potential of −−70 mV to 0 mV in PV interneuron (fast inward Na+ currents were partially removed), which can be blocked by gabazine (gray). Note the expected fixed latency for all single-trial responses. *Lower panel*. Example trace

showing a failure in autaptic response. Scale: 100 pA, 5 ms. **B**. Cuprizone-treated mice show a significant decrease in the percentage of cells showing an autaptic response (*Fisher's exact test; \*p = 0.0173*). **C, D**. There was no difference between groups in the amplitude (**C**) or decay (**D**) of the autaptic post-synaptic currents in cells that had autapses (*t test; n = 40/28 cells per group, p = 0.1246 (C) and p = 0.9350 (D)*). **E**. An example image of a filled PV interneuron showing putative autapses with white arrows. Note that all autapses are present in close distance to the soma. **F**. Pie charts depicting the percentage of cells showing (1) both structural and physiological autapses, (2) no structural nor physiological autapses, and (3) only structural autapses. Note that only mice that underwent juvenile demyelination have PV interneurons which show structural autapses but no autaptic response. **G**. Representative voltage steps showing an AutIPSC response in black and no response in red. The red trace shows a cell that had NO autaptic response. Scale:100 pA, 5 ms. **H**. Shiverer mice show a clear decrease in the percentage of cells showing an autaptic response compared to control mice (*Fisher's exact test; p = 0.0516*). **I, J**. There was no difference between groups in the amplitude (I) or decay (J) of the autaptic post-synaptic currents in cells that had autapses in the Shiverer mice (light red) compared to the control mice (black) (*t test; n = 27/29 cells from 9 mice per group, p = 0.7167 (I) and p = 0.2377 (J)*). **K**. Representative voltage steps showing an AutIPSC response in control (black) and adult demyelination (blue) groups. The red trace shows a cell that had NO autaptic response. Scale: 50 pA, 2 ms. **L**. Adult demyelination mice show no difference in the percentage of cells showing an autaptic response compared to control mice (*Fisher's exact test; p = 0.7488*). **M**. There was a significant decrease in the averaged amplitude of AutIPSCs recorded in mice that underwent adult demyelination (*t test; n = 26/18 cells from 8 to 6 mice per group, \*p = 0.0194*). **N**. There was no difference in the decay of the autaptic post-synaptic currents (*t test; n = 26/18 cells from 8−6 mice per group, p = 0.7804*). The data displayed in (C–D), (I–J), and (M–N) can be found in S7 Table.

response compared to only 11.7% (2 out of 17) in the control group and in the adult demyelination group only 10.0% (1 out of 10) of the cells showed failure in autaptic responses at 200 Hz (Fig 8G and 8H).

**Loss of PV interneuron autapses correlates with high-frequency firing impairments.** Juvenile demyelination causes impairment in high-frequency firing of PV interneurons and a loss of functional autapses, however, is there a link between these two phenotypes? To explore this further, we divided the cells from both the juvenile demyelination group and the control group into (1) cells with an autaptic response and (2) cells with *no* autaptic response (S5 Fig).

In the control group, we found that PV interneurons with an autaptic response were less excitable than cells with no autaptic response, with a difference in rheobase of ~40 pA (S5A and S5B Fig). PV interneurons with autaptic response also showed an increased AHP without a change in AP width (S5C and S5D Fig). Importantly, although cells with no autaptic response had an increased firing rate following low-current injections (0–250 pA), their firing rate was attenuated in response to high-current injections (300–600 pA) compared to cells with autaptic response (S5E and S5F Fig). These data indicate that autapses are necessary but not sufficient for high-frequency firing in PV interneurons.

In mice that underwent juvenile demyelination, 60.0% (18 out of 30) of PV interneurons that had no autaptic response showed impaired firing at high frequencies (100–250 Hz). Conversely, 60.0% (12 out of 20) of PV interneurons with impaired firing at high frequencies had no autaptic response (S5G and S5H Fig), suggesting that the absence of autapses in the juvenile demyelination group increased the likelihood of PV interneurons exhibiting impaired firing. Indeed, 31.0% (16 out of 52) of all PV interneurons from the juvenile demyelination group showed both no autaptic responses and an impaired firing phenotype versus only 4.6% (2 out of 43) in the control group. While these results indicate that the relationship between the loss of functional autapses and the impairment in high-frequency firing in the juvenile demyelination group may not be causal, they indicate that a loss of autapses during the critical period of development of PV interneurons might facilitate their probability of failure to fire at higher frequencies.

**Remyelination in adulthood only partially restores PV interneuron properties in the PFC.** Our data reveal a clear impairment in PV interneuron properties following juvenile demyelination. We next questioned whether remyelination could restore PV interneuron function in PFC. To that end, mice underwent juvenile demyelination from P21 to P60, after which they returned to a normal diet from P60 to P100 for remyelination purposes (Fig 9A–9C). No difference was detected in the overall myelin expression in PFC after remyelination (*Fisher's exact test; p = 0.509;* Fig 9B). We then performed whole-cell recordings from PV interneurons in the PFC. Both the input resistance and the sag amplitude were restored to control levels upon remyelination (Fig 9D–9G). Interestingly, we found that remyelination caused a significant increase in the rheobase of PV interneurons, promoting a decreased excitability of these cells (Fig 9H and 9I). When examining the AP waveform, we detected a small increase in the decay of the AP in the

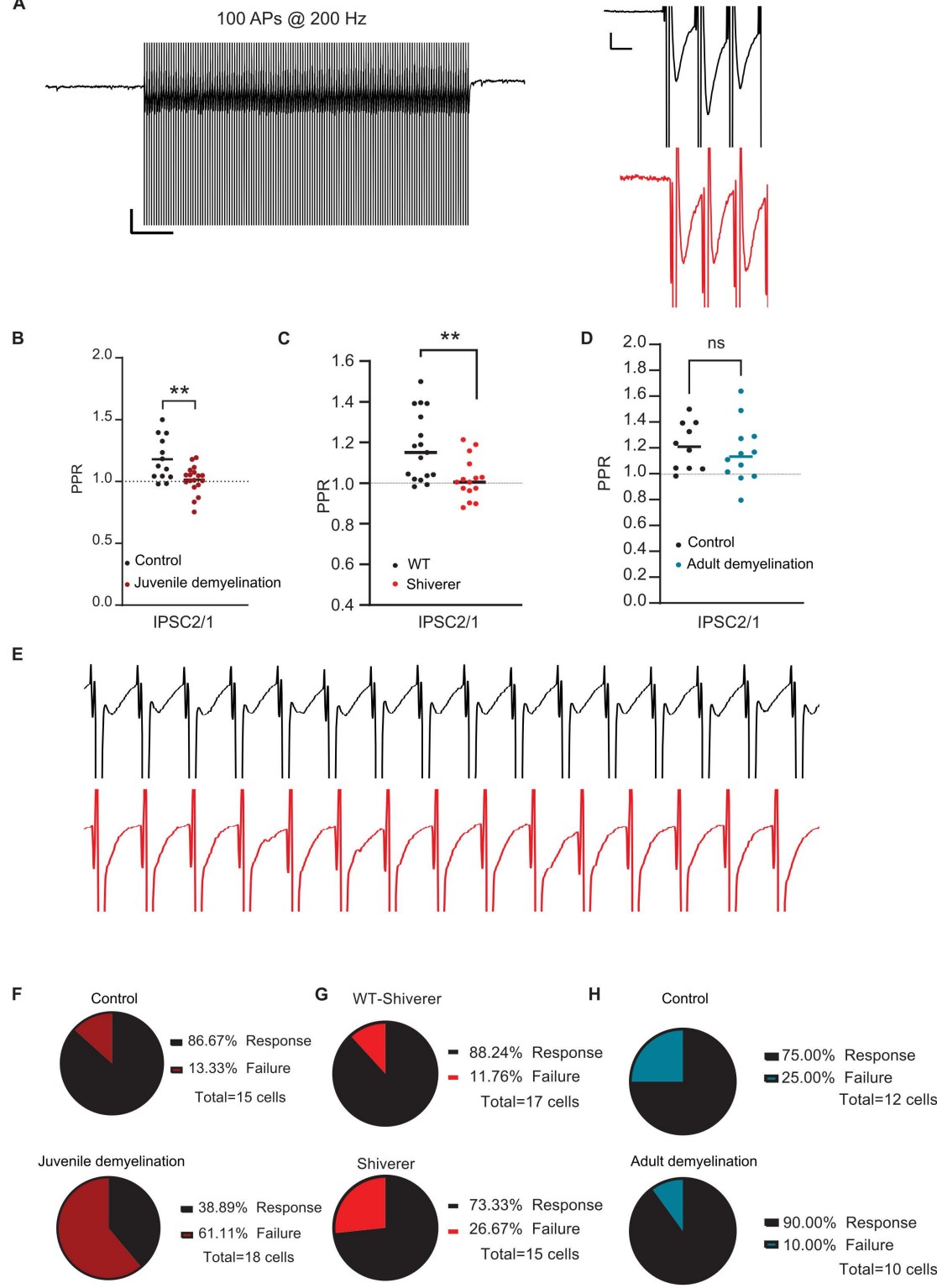

**Fig 8. Juvenile demyelination impairs autaptic plasticity. A.** *Left.* Example trace of the train of 100 AP at 200 Hz in voltage-clamp mode. *Right.* Inset of the first three unitary autaptic IPSCs in a train of 100 evoked at 200 Hz in PFC PV interneurons from control mice (black) and cuprizone-treated mice (red); Note the clear facilitation in the second response in the control mice. **B**. Quantification of the paired-pulse ratio (PPR) of both control (black

dots) and cuprizone-treated (red dots) mice reveal an impairment in short-term plasticity at autaptic site following juvenile demyelination (*t test: n = 13,18 cells from 7 to 8 mice per group, **p = 0.0072*). **C**. Short-term plasticity of autaptic responses at 200 Hz is impaired in shiverer mice (light red) compared to control (black) mice (*n = 17/15 cells from 9 mice per group, **p = 0.0041 (IPSC2/1)*). **D**. No difference was detected in the paired-pulse ratio (PPR) between control (black dots) and adult demyelination (blue dots) (*n = 12,10 cells from 8 to 6 mice per group, p = 0.5780 (IPSC2/1)*). **E**. Example traces of the first 15 unitary autaptic IPSCs in a train of 100 evoked at 200 Hz in PFC PV interneurons from control mice (black) and cuprizone-treated mice (red); Note that many evoked responses failed in the cuprizone mice. **F**. Cuprizone-treated mice show a significant increase in the percentage of cells showing a failed autaptic response at 200 Hz (*Fisher's exact test; *p = 0.0110*). **G**. There was no significant difference in the percentage of cells showing a failed autaptic response at 200 Hz remains in the Shiverer group compared to the control group (*Fisher's exact test; p = 0.3828*). **H**. There was no difference in the percentage of cells showing a failed autaptic response at 200 Hz remains in the adult demyelination group compared to the control group (*Fisher's exact test; *p = 0.5940*). The data displayed in (B), (C), and (D) can be found in S8 Table.

remyelination group, even though AP half-width was not significantly different (Fig 9J–9N). PV interneurons from the remyelination group still showed decreased firing frequency in response to increased current injections (Fig 9O–9Q). Even after remyelination, 26.2% (11 out of 42) of the cells still could not sustain their firing at high frequency, compared to 5.0% (1 out of 20) in control mice (Fisher's Exact Test, $p = 0.083$; Fig 9R and 9S). We then investigated whether self-inhibitory transmission was restored by remyelination (Fig 9T–9Y). Remyelination resulted in a rescue of autaptic neurotransmission (autaptic responses: remyelination, 68.4%, 19 out of 28; control, 75.0%, 15 out of 20; *Fisher's Exact Test, p = 0.764*;) (Fig 9T and 9U). The amplitude and decay of the responses were also similar across groups (Fig 9V and 9W). No significant difference between groups was detected in short-term plasticity at 200 Hz (IPSC2/1; mean PPR: 1.18 for remyelination group and 1.12 for control group) (Fig 9X). Yet, there was a significant difference in the percentage of cells that showed failures at 200 Hz in the remyelination group (Fisher's Exact Test, *$p = 0.023$; Fig 9Y). These results suggest that remyelination leads to a partial rescue of PFC PV interneurons, in which the impairment remains apparent when PV interneurons are driven to fire at high frequencies.

## Discussion

### Myelination of PFC PV interneurons during adolescence is crucial for their structural and electrophysiological maturation

Several studies have highlighted the importance of adult PV interneuron myelination in proper PV interneuron function, such as inhibitory control of local oscillations, experience-dependent plasticity, or AP conduction velocity increase along PV axons [6,8,25]. Yet, little is known about whether PV interneuron myelination is important for cell-autonomous development and maturation. Given the implications of PFC development in many neuropsychiatric disorders, along with a clear role of PV interneuron dysfunction in such disorders, adding to it the various reports of myelin deficits in patients with neuropsychiatric disorders [9], we aimed to investigate whether an early disruption of PV interneuron myelination, during the critical period of PFC development, can lead to an impairment in cell autonomous maturation and development. Many studies report an extended critical period for PFC maturation from P21 to P35 [13,16,26,27]. We used the cuprizone model starting at P21 to induce juvenile demyelination. Our data revealed that juvenile demyelination leads to persistent alterations in PV interneuron morphology in adulthood. Specifically, we found a decrease in total axonal length, as well as the number of axonal branches among PV interneurons from mice that underwent juvenile demyelination. Axons of PV interneurons have previously been reported to have considerable morphological plasticity during the second and third postnatal week, at a time where myelination is starting to peak [3,28,29].

To what extent might these structural changes be related to PV interneuron intrinsic properties and electrophysiological development? Our data revealed that PV interneurons in the PFC of mice that underwent juvenile demyelination show clear changes in their intrinsic properties. Specifically, we observed an increase in the input resistance and sag amplitude, an increase in AP width and a substantial decrease in the maximum firing frequency, all reminiscent of immature PV interneurons [1,19]. Notably, among mice that underwent juvenile demyelination, ~40% of the

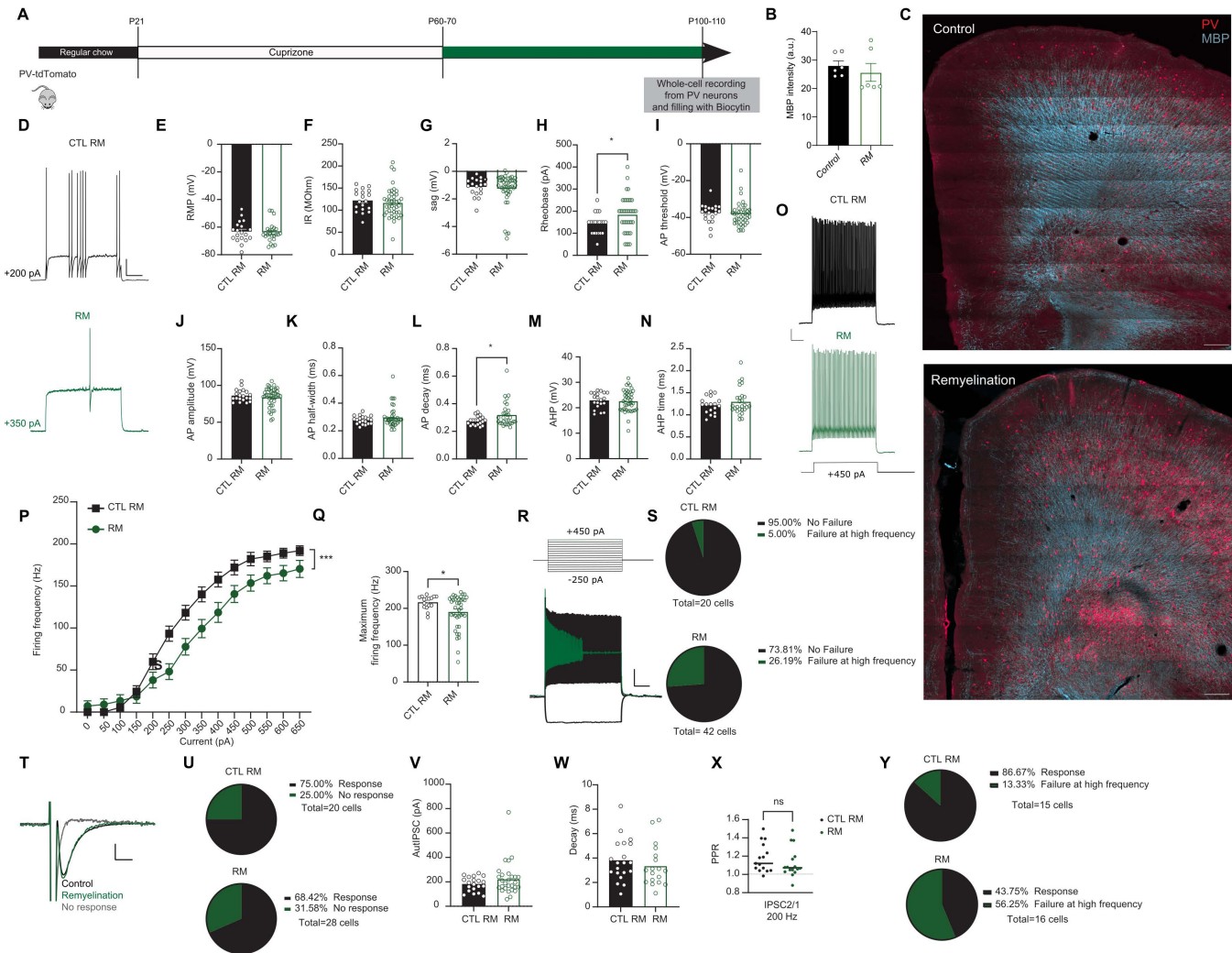

**Fig 9. Remyelination in adulthood leads to an incomplete restoration of PV interneuron properties. A.** Experimental design for remyelination in PV-tdTomato mice **B**. Quantification of the overall myelin intensity in PFC of both groups shows no difference in overall myelin expression in the remyelination compared to the control group (*t test; n = 3 mice per group, p = 0.509). **C.** *Upper panel.* Confocal overview image of PFC in PV-tdTomato animal (tdTomato+, red) overlaid with myelin basic protein (MBP, cyan). *Lower panel.* Confocal overview image immunolabeled for MBP expression showing remyelination of the PFC region following 5 weeks of normal diet. Scale: 300 μm. **D**. Representative traces of voltage responses following a depolarizing step from control (black) and remyelination (green) mice illustrating an increase in Rheobase in mice after remyelination. Scale: 20 mV, 100 ms. **E–I**. Summary data showing the averaged (± s.e.m) of the following intrinsic properties: **(E)** Resting membrane potential (RMP) (*t test; n = 25/19 cells from 8 mice per group, p = 0.7435*), **(F)** Input resistance (IR) (*t test; n = 25/19 cells from 8 mice per group, p = 0.4626*), **(G)** Sag (*t test; n = 25/19 cells from 8 mice per group, p = 0.6088*), **(H)** Rheobase (*t test; n = 25/19 cells from 8 mice per group, \*p < 0.05*) and **(I)** Action potential (AP) threshold (*t test; n = 25/19 cells from 8 mice per group, p = 0.6472*). **J–N**. Summary data showing the averaged (± s.e.m) of the following AP waveform properties: **(J)** AP amplitude (*t test; n = 25/19 cells from 8 mice per group, p = 0. 23234*), **(K)** AP half-width (*t test; n = 25/19 cells from 8 mice per group, p = 0.2202*), **(L)** AP decay (*t test; n = 25/19 cells from 8 mice per group, \*p < 0.05*), **(M)** After-hyperpolarization (AHP) amplitude (*t test; n = 25/19 cells from 8 mice per group, p = 0.8156*) and **(N)** AHP time (*t test; n = 25/19 cells from 8 mice per group, p = 0.3123*). **O**. Representative traces of voltage responses following +450 pA current injection. Scale bar: 100 ms, 10 mV. **P**. Average action potential (AP) frequency in response to 0-650 pA current steps illustrating a significant decrease in PV interneuron firing frequency in remyelination (green) mice (*group x current two-way repeated measures: n = 25/19 cells from 8 mice per group: F(13,546) = 3.962, \*\*\*p < 0.001*). **Q**. Summary data of the maximum firing frequency per group (*t test; n = 25/19 cells from 8 mice per group, \*p < 0.05*). **R**. Example trace of a PV interneuron that is unable to maintain high-frequency firing at increased current injection. **S**. Percentage of cells that failed to maintain high frequency firing at > 500 pA is still not fully recovered after remyelination (*Fisher's exact test; p = 0.0828*). **T**. Representative voltage steps showing an AutIPSC response in control (black) and remyelination (green) groups. The red trace shows a cell that had NO autaptic response. Scale::50 pA, 2 ms. **U**. Remyelination mice show no difference in the percentage of cells showing an autaptic response compared to control mice (*Fisher's exact test; p = 0.7639*). **V, W**. There was no difference between groups in the amplitude (V) or decay (W) of the autaptic post-synaptic currents in cells that had autapses in the remyelination group (green) compared to the control group (black) (*t test; n = 28/20 cells from 8 mice per group, p = 0.1939*

*(V) and n = 20/18 cells from 8 mice per group, p = 0.4328 W).* **X, Y**. *Paired-pulse ratio (PPR) of both control (black dots) and remyelination (green dots) mice shows no difference in short-term plasticity at autaptic sites specifically at 200 Hz (t test; n = 15/16 cells from 8 mice per group, p = 0.3388). Yet, the percentage of cells showing a failed autaptic response at 200 Hz remains increased in the remyelination group compared to the control group (Fisher's exact test; \*p = 0.0233). The data displayed in (B), (E–N), and (D) can be found in* S9 Table.

PV interneurons were unable to sustain high-frequency firing, for which their maximum firing rate was substantially decreased. Similar observations were made in the *shiverer* model, suggesting that the lack of myelin during this critical period of development had a significant impact on the electrophysiological maturation of these cells. Indeed, in mice that underwent adult demyelination, high-frequency firing was not affected. Taken together, our results indicate that the ontogeny of myelination is crucial for the proper structural and physiological development of PV interneuron.

## Kv3 is crucial for PV maturation and myelination

PV interneuron electrophysiological maturation has been closely linked to the development of distinct ion channels that allow for their unique ability to fire at high frequencies. Specifically, Kv3 channels have been shown to double their expression level from P18 to P30 [1]. Therefore, we next explored whether a loss of myelin during adolescence could be affecting Kv3 expression and activity levels in PV interneurons of the PFC. Previous studies have demonstrated that myelination causes alterations in Na+ and K+ channels in pyramidal neurons [30]. Our results confirmed a clear decrease in Kv3.1 expression by immunofluorescence, and K+ current using voltage-clamp recordings from PV interneurons of the PFC in mice that underwent juvenile demyelination; this could explain the increased AP width and impaired firing that we observed. We next tested whether AUT00201, a positive modulator of Kv3 K+ currents, could restore Kv3 currents in PV interneurons from mice that underwent juvenile demyelination. Bath application of AUT00201 rescued the deficit in AP width but not the alteration of high-frequency firing. This indicates that, while acute Kv3 activation can restore AP waveforms in these cells, the loss of Kv3 expression due to juvenile demyelination likely requires a more chronic intervention to fully rescue PV maturation. Moreover, it is not unlikely that other channels might also be affected by juvenile demyelination. In particular, the changes in the sag we observed following juvenile demyelination and in *shiverer* mice are reminiscent of alterations in HCN channel function. HCN channels have been reported to be exclusively localized on axons in PV interneurons, and play a critical role in enhancing AP initiation during sustained high-frequency firing and in increasing the velocity by which APs propagate along PV interneuron axons [31]. Moreover, expression of TASK-1 and -3 channels, which are critical for maintaining the RMP and modulation of the excitability of PV interneurons, increases during adolescence, for which PV interneurons from TASK-1/3 double knockout mice exhibit increased input resistance, wider AP, and a decreased firing rate [32]. More research is needed to explore the mechanism by which demyelination during the juvenile period could decrease the expression of Kv3 and its function. The loss of myelin causes highly significant changes in nodes of Ranvier [33,34] and demyelination could have disrupted the localization of Kv3 along the soma and axon of PV interneurons. Moreover, previous studies have found that loss of oligodendrocyte support and axon-glia signaling, which occurs in cuprizone-induced demyelination, can lead to a decreased expression of certain potassium channels (e.g., Kv3) due to altered neuronal activity or disrupted transcriptional support [30,35]. Importantly, there might be a reciprocal link between Kv3 function and PV interneuron inability to fire at high frequencies by which demyelination could be contributing to an increased leakage of potassium currents and subsequent reduction in AP fidelity and conduction failures [34,36]. The decrease in Kv3 currents could also be a result of loss of metabolic support from oligodendrocytes and altered axonal homeostasis and ion balance [35,37]. More studies are required to explore the specific mechanisms involved in such a decrease in Kv3 function following demyelination, and how this links to PV interneurons failure to fire at high frequencies.

While the observed decrease in Kv3 current is highly relevant in the context of PV interneuron maturation, especially given the vast literature about the role of Kv3 in sustaining high-frequency firing, it is important to note that the pharmacological agent used, tetraethylammonium (TEA), lacks high selectivity for KV3 channels [38], and the use of fluorescence intensity of ion channels is not ideal for making definitive claims about expression changes. The aim with the immunofluorescence was to provide supporting evidence to the electrophysiological recordings. This makes the interpretation of our findings related to KV3 function more complex and less definitive. As discussed above, other channels are likely to be affected as well, and the current data cannot describe which channels were affected by juvenile demyelination.

## Myelination of PFC PV interneurons during adolescence is important for self-inhibitory transmission

What happens to autapses following juvenile demyelination and the consequent aberrant development of PV interneurons? Our data revealed a significant decrease in the number of PV interneurons that have autaptic responses after juvenile demyelination, but not after adult demyelination. In examining reconstructed neurons, we found that ~25% of PV interneurons from the juvenile demyelination group exhibited structural autapses but no autaptic response, indicating that the loss of myelin during adolescence led to an impairment of autaptic neurotransmission. Moreover, ~ 30% of the cells had no discernible autapses, compared to only 11% in controls, suggesting a complete loss of autapses in some PV interneurons due to juvenile demyelination. Yet, more than 45% of PV interneurons of mice that underwent juvenile demyelination still had functional autapses, in which the amplitude of the responses appeared largely unaffected. Therefore, we next investigated whether plasticity at autaptic sites was impaired as a result of juvenile demyelination. Our data uncovered an impairment in short-term plasticity at autaptic sites at 200 Hz, indicating that myelination during adolescence is required for adult synaptic plasticity at high frequencies. Results in *shiverer* PV interneurons confirmed these data. Conversely, in mice that underwent adult demyelination, short-term plasticity at autaptic sites was not affected, although, the amplitude of the inhibitory autaptic response was decreased. This is likely due to a deficit in inhibitory synapse maintenance caused by the loss of myelin, as previously reported [8]. Together, our data suggest that juvenile demyelination can cause a loss of functional autapses in PV interneurons of the PFC and impair autaptic neurotransmission. It remains to be investigated whether the deficits in short-term plasticity recorded in PV interneurons from mice that underwent juvenile demyelination are due to failures in AP-triggered GABA release or failures of AP propagation along demyelinated axons.

## Autapses are associated with high-frequency firing of PV interneurons

Previous studies have demonstrated that fast-spiking interneurons exhibit autapses, which are synaptic contacts between the cell's own soma or dendrite and its own axon [3,22–24,39]. While there is still a lot unknown about the role of these autapses, it has been proposed that they play an important role in the temporal control of microcircuits by modulating PV interneuron excitability. Specifically, autaptic neurotransmission has been proposed as a particular form of cortical disinhibition regulating PV interneuron influences on network oscillations [22,24]. Approximately 70%–80% of PV interneurons have autapses, while non-fast-spiking interneurons do not exhibit autapses [23,24,39,40,]. PV interneuron autapses are GABAergic with high probabilities of release, in which plasticity appears to follow largely similar rules as non-autaptic inhibitory postsynaptic sites [22,23,39,40]. Interestingly, a recent study found that autaptic neurotransmission induced larger responses than other synaptic inputs in a subset of PV interneurons, and they estimated that autaptic transmission accounted for approximately 40% of the global inhibition that PV interneurons received [22].

In the current study, we aimed to explore whether autapses are associated with the ability of PV interneuron to fire at high frequency, which in turn has been demonstrated to be of importance for network oscillations and higher cognitive processing [22,24]. We show that PV interneurons without autapses are more excitable, displaying a lower rheobase and a change in the AP's AHP. Interestingly, these cells had higher firing frequencies at low current injections, yet they had a

significantly decreased maximum firing rate. These data suggest that autapses contribute to PV interneuron's ability to fire at high frequency by controlling the excitability and the timing of PV interneurons spiking activity, as proposed in previous studies [23,24,41]. It is important to note that while associations between demyelination and deficits in high-frequency firing were observed, we cannot definitively conclude causal relationships without more targeted experimental approaches.

## Remyelination after juvenile demyelination is unable to fully rescue PV interneuron function

Remyelination studies have revealed that in only 4 weeks after prolonged cuprizone treatment, nearly complete remyelination can occur [42–45]. We therefore investigated whether remyelination could restore PV interneuron properties and autaptic neurotransmission following juvenile demyelination. Our data showed that while many of the membrane properties were rescued by remyelination, PV interneuron firing rate was still decreased after remyelination and more than 25% of the cells could not sustain their firing rate at high frequencies. Interestingly, autaptic responses at high frequency were also still impaired. This suggests that restoring myelin in the PFC might have reinstated some channel properties in PV interneurons, yet the ability of PV interneurons to fire at high frequency was not fully restored. We propose that the loss of myelin during this critical period of PFC development constricts any full rescue of PV interneuron properties. This is in line with many previous works that have confirmed that this is a sensitive period for PFC development and any intervention during this vulnerable period could have long-lasting effects [12,16,27,46]. It is important to note that accurate quantification of myelination remains technically challenging, and although we used established approaches, limitations in spatial resolution and sensitivity means that we can't fully rule out that there was no full remyelination of PV interneurons axons in our remyelination group. Other studies have indeed showed that even though upon cuprizone withdrawal, significant remyelination is observed, particularly in regions like the corpus callosum and cerebellar peduncles, the recovery is less efficient after prolonged exposure (more than 8 weeks) [47,48]. Moreover, and despite structural remyelination, behavioral deficits and continued axonal degeneration can persist, indicating that remyelination does not fully restore function [49,50]. This limitation is especially relevant given recent reports showing that remyelination is often incomplete and does not restore original myelination patterns [51].

## Relevance for human disorders

The development of PFC circuits is known to extend beyond the juvenile developmental period [14,17]. In rodents, it has been characterized by an essential maturation of inhibitory networks and the establishment of excitatory/inhibitory balance, which in turn give rise to various brain oscillations involved in higher cognitive functions [17,52]. PV interneuron development follows a prolonged period of maturation that also extends into adolescence [1,9,19]. While many studies have attributed an impairment in PFC development and PV interneuron maturation to many neuropsychiatric disorders [11], little is known about whether there is a link between myelination and PV interneuron development in the PFC. Makinodan and colleagues were the first to identify a critical period for oligodendrocyte maturation and myelination for PFC-dependent social behavior in rodents [16]. They showed that social isolation of mice between P21 and P35 can have long-lasting effects on PFC myelination, where mice that underwent social isolation in this period showed less myelin expression in the PFC compared to mice that were socially isolated after P35 and control mice. Furthermore, mice in which myelination was impaired because of loss of ErbB3 signaling showed similar social behavior deficits to the ones elicited by social isolation during the critical period, confirming that myelination of the PFC during this period is necessary for normal social behavior in the adult. On the other hand, PV interneurons have been found to be essential for regulating network synchrony and microcircuit precision, making their involvement in learning and memory unsurprising. Disrupting PV neuron function—either by removing them from hippocampal CA1 or impairing their excitatory inputs—leads to deficits in spatial working memory and network integrity [53–55]. They are also key players in associative memory across brain regions [53,56].

Inhibiting PV neurons in the hippocampus or PFC during post-learning sleep impairs fear memory consolidation, likely through disrupted ripple–spindle coupling between hippocampus and cortex [57–59]. Notably, such inhibition after

consolidation or during retrieval has no effect, highlighting a time-sensitive role. By controlling excitatory neuron recruitment, PV interneurons help define memory engram composition and prevent nonspecific activation [53,60]. Their silencing increases engram size and alters Arc gene expression, suggesting a role in memory precision [61]. PV neurons also regulate place cell tuning and maintain spatial representations [62]. Overall, PV interneurons support memory consolidation by shaping ensemble activity, maintaining oscillatory coordination, and preserving context specificity. Taken together, we propose that our data highlights an important link between PV interneuron myelination and their development in the PFC during this critical period, which offers key insights into the mechanisms underlying the pathophysiology and potential therapies for neurodevelopmental disorders.

## Limitation of the study

While this study unravels important findings about the role of juvenile myelination in PV interneuron maturation and physiology, it also has several limitations that should be considered when interpreting the results. First, sex differences were not assessed due to the limited sample size. This restricts our ability to determine whether the observed effects are consistent across sexes or potentially influenced by sex-specific factors. Second, the lack of cell-specific demyelination introduces uncertainty regarding the mechanistic underpinnings of the observed changes. It remains unclear whether these alterations are predominantly due to demyelination PV interneuron axons, excitatory axons, or a combination of both. Further work employing cell-type-specific manipulation is needed to resolve this ambiguity. On that note, the model used is based on systemic administration of cuprizone via the diet of mice. Known peripheral effects, particularly on hepatic function, could lead to secondary metabolic changes that may influence central nervous system outcomes [47]. Hence, although the cuprizone model is widely used to study demyelination and remyelination, it has shortcomings. Spontaneous remyelination can occur even during continued cuprizone exposure, and behavioral deficits may persist long after histological recovery [47]. In this context, the observed downregulation of Kv3 channels may reflect a correlative rather than causative relationship with myelin loss. This highlights the need for caution when linking cellular changes to functional outcomes.

## Materials and methods

All experiments were conducted under the approval of the Dutch Ethical Committee (AVD1010020173544) and in accordance with the Institutional Animal Care and Use Committee (IACUC) guidelines. The Pv-tdTomato [Tg(Pvalb-tdTomato)15Gfng] line was used as well as the Shiverer [B6;C3Fe.SWV-Mbpshi/J (Shiv) (Chernoff 1981; www.jax.org/strain/001428)] line that was crossed with the Pv-tdTomato line.

For cuprizone treatment, mice received 0.2% (w/w) cuprizone (Bis(cyclohexanone)oxaldihydrazone (C9012, Merck) added to grinded powder food or to food pellets (Envigo). For the juvenile demyelination group, mice received fresh cuprizone food starting at P21 for a period of 6–7 weeks while control mice received control food, both *ad libitum*. For the adult demyelination group, mice received either cuprizone food (cuprizone group) or normal chow food (control group) starting P60 for a period of 6–7 weeks. The average maximum weight loss during cuprizone treatment was around 15% in the adult cuprizone group and around 7% in the juvenile demyelination group. All mice from both sexes were used for these experiments. All mice were maintained on a regular 12 h light/dark cycle at 22 °C (±2 °C) with access to food and water *ad libitum*. All mice were group-housed to rule out the effects of social isolation on control and cuprizone mice.

## Slice preparation and electrophysiology in mouse ex vivo slices

Whole-cell recordings of PV interneurons in layers 2–3 in the PFC were performed as described previously [5,63]. Briefly, after decapitation, brains were placed in ice-cold partial sucrose-based solution containing (in mM): sucrose 70, NaCl 70, NaHCO$_3$ 25, KCl 2.5, NaH$_2$PO$_4$, 1.25, CaCl$_2$ 1, MgSO$_4$ 5, sodium ascorbate 1, sodium pyruvate 3, and D(+)-glucose 25 (carboxygenated with 5% CO$_2$/95% O$_2$). Coronal slices from the PFC (300 μm thick) were obtained with a vibrating slicer (Microm HM 650V, Thermo Scientific) and incubated for 45 min at 34 °C in holding artificial cerebrospinal fluid (ACSF)

containing (in mM): 127 NaCl, 25 NaHCO$_3$, 25 D(+)-glucose, 2.5 KCl, 1.25 NaH$_2$PO$_4$, 1.5 MgSO$_4$, 1.6 CaCl$_2$, 3 sodium pyruvate, 1 sodium ascorbate, and 1 MgCl$_2$ (carboxygenated with 5% CO$_2$/95% O$_2$). Next, the slices recovered at room temperature for another 15 min. Slices were then transferred into the recording chamber where they were continuously perfused with recording ACSF (in mM): 127 NaCl, 25 NaHCO$_3$, 25 D-glucose, 2.5 KCl, 1.25 NaH$_2$PO$_4$, 1.5 MgSO$_4$, and 1.6 CaCl$_2$. Cells were visualized using an upright microscope (BX51WI, Olympus Nederland) equipped with oblique illumination optics (WI-OBCD; numerical aperture 0.8) and a 40× water-immersion objective. Images were collected by a CCD camera (CoolSMAP EZ, Photometrics) regulated by Prairie View Imaging software (Bruker). Layer II–III pyramidal cells in the somatosensory cortex were identifiable by their location and morphology. Electrophysiological recordings were acquired using HEKA EPC10 quattro amplifiers and Patchmaster software (10 Hz sampling rate) at 33°C. Patch pipettes were pulled from borosilicate glass (Warner instruments) with an open tip of 3.5–5 MegaOhm of resistance and filled with intracellular solution containing (in mM) 125 K-gluconate, 10 NaCl, 2 Mg-ATP, 0.2 EGTA, 0.3 Na-GTP, 10 HEPES, and 10 K2-phosphocreatine, pH 7.4, adjusted with KOH (280 mOsmol/kg), with 5 mg/mL biocytin to fill the cells. Series resistance was kept under 20 M with correct bridge balance and capacitance fully compensated; cells that exceeded this value were not included in the study. Cells were filled with biocytin for at least 20 min. AMPA-mediated currents were blocked using DNQX (10 μM, HelloBio). Gabazine (10 μM, HelloBio) was used to block GABAARs. AutIPSC recordings were measured as described previously. For Kv current recordings, holding potential was −70 mV and voltage steps from −70 mV to +60 mV (in 10 mV increments) were recorded to extract high voltage-activated potassium currents. Tetrodotoxin (1 μM) was used to suppress sodium-activated channels. TEA (1 mM) was applied to the bath following baseline recordings. The resulting voltage currents were obtained by subtraction of baseline from TEA-sensitive currents. Leak currents were removed online.

Intrinsic passive and active membrane properties were recorded in current-clamp mode, both at resting membrane potential and at −70 mV, by injecting 500 ms of increasing current stimuli from −300 pA to +650 pA, at intervals of 50 pA. Data analysis was conducted using a custom-designed script in Igor Pro-9.0 (Wavemetrics).

## AUT00201 compound

AUT00201 (Autifony Therapeutics) is a small molecule positive allosteric modulator of Kv3.1 and Kv3.2 channels. AUT00201 (10 mM stock solution) was dissolved in DMSO (0.1%), and the final concentration used in the recording chamber was 1 μM.

## Immunofluorescence

Mice were anaesthetized intraperitoneally with pentobarbital natrium (Nembutol) before performance of cardial perfusion using 4% paraformaldehyde (PFA). Brains were removed and fixed in 4% PFA for 2 h at room temperature. Next, brains were transferred into 10% sucrose phosphate buffer (0.1 M PB) and stored overnight at 4 °C. To improve the slicing process, brains were embedded in 12% gelatin-10% sucrose blocks and left during 1.5 h at room temperature in 10% paraformaldehyde-30% sucrose solution (PB 0.1 M) and later incubated overnight in 30% sucrose solution at 4 °C. Brains were then sliced in coronal slices at 40 μm thick using a freezing microtome (Leica, Wetzlar, Germany; SM 2000R) and stored in 0.1 M PB. For the staining, slices were blocked with 0.5% Triton X-100% (MerkMillipore) and 10% normal horse serum (NHS; Invitrogen, Bleiswijk, The Netherlands) for 1 h at room temperature and incubated over 72 h at 4 °C with primary antibodies mouse anti-Kv3.1 (1:300, Synaptic systems, Cat. No.: 242 003), rabbit anti-PV (1:1,000, Swant PV25), and goat anti-MBP (1:300, Santa Cruz, C-16, sc-13 914) in phosphate-buffered saline (PBS) buffer containing 0.4% Triton X-100 and 2% NHS. Secondary antibodies anti-mouse Alexa 488 (1:300, Invitrogen), anti-rabbit Cy3 (1:300, Invitrogen), and anti-rabbit Alexa 647 (1:300, Invitrogen) were employed for 2 h at room temperature. Slices were coverslipped with Vectashield H1000 fluorescent mounting medium (Vector Labs, Peterborough, UK).

Recovery of biocytin-labeled cells following electrophysiological recordings was performed as reported before [64]. Patch-clamp recorded PV interneurons were filled with 5 mg/mL biocytin during whole-cell recordings and then fixed with 4% (PFA) overnight and stored in PBS at 4 °C. Specifically, 300 µm slices were incubated overnight at 4°C in fresh 4% PFA. Slices were extensively rinsed at room temperature in PBS and stained in PBS buffer containing 0.4% Triton X-100, 2% normal horse serum (NHS; Invitrogen, Bleiswijk, the Netherlands) and streptavidin-Cy3 (1:300; Invitrogen) overnight at 4°C. Slices were washed with PBS and PB 0.1 M and mounted on slides, cover slipped with 150 µl Vectashield, sealed, and imaged for their axonal morphology (see Confocal imaging and Reconstruction). To avoid excessive thinning or dehydration of 300 µm sections, cells were mounted, immediately imaged and returned to PB 0.1 M directly after imaging.

After full cell imaging, 300 µm slices were extensively washed in PB 0.1 M and incubated overnight at 4°C in 30% sucrose (0.1 M PB). Sections were then carefully recut at 40 µm using a freezing microtome (Leica, Wetzlar, Germany; SM 2000R) and stored serially in 0.1 M PB at 4°C. Serial 40 µm sections were extensively washed with PBS and pre-incubated with a blocking PBS buffer containing 0.5% Triton X-100% and 10% NHS for 1 hr at room temperature. Sections were incubated in PBS buffer containing 0.4% Triton X-100% and 2% NHS for 72 hr at 4°C and goat anti-MBP. Then, sections were washed with PBS and incubated with corresponding Alexa-conjugated secondary antibodies (1:300, Invitrogen) in PBS buffer containing 0.4% Triton X-100, 2% NHS for 5 hr at room temperature. Sections were washed with PB 0.1M and mounted on slides, cover slipped with Vectashield H1000 fluorescent mounting medium (Vector Labs, Peterborough, UK), sealed and imaged. The following primary antibodies were used: mouse anti-PV (1:1000, Swant, 235), and goat anti-MBP (1:300, Santa Cruz, C-16, sc-13,914).

## Confocal imaging

As previously described ([5]; [64]), images were acquired using a Zeiss LSM 700 confocal microscope (Carl Zeiss) equipped with Plan-Apochromat objectives: 10×/0.45 NA, 40×/1.3 NA (oil immersion), and 63×/1.4 NA (oil immersion). Alexa Fluor 488 and Cy3-conjugated secondary antibodies were visualized using excitation wavelengths of 488 nm and 555 nm, respectively. A 10× objective was used to capture low-magnification images, and then a high-resolution whole-cell images were acquired with the 63× oil immersion objective using the 555 nm excitation wavelength. Imaging parameters were standardized, with the pinhole set to 0.2% and gain adjusted between 750 and 800 units to optimize signal-to-noise ratio.

Biocytin-filled cells were imaged using tiled z-stacks (512 × 512 pixels) with a step size of 1 µm. Resectioned slices of these cells were imaged at 40× magnification using both 488 and 555 nm excitation wavelengths. Myelin basic protein (MBP) staining was visualized at 40× magnification (1,024 × 1,024 pixels) with 555 nm excitation.

All imaging settings were kept consistent across samples to ensure uniform fluorescence quantification.

For Kv3.1b immunofluorescence analysis, confocal pictures of PV and Kv3.1 stainings were taken using 40×/1.3 NA (oil immersion) objective. 20× pictures were used for quantification of cell numbers, and 40× z-stack confocal picture were used for fluorescence analysis. Kv3 intensity was quantified as previously described [36]. Four stacks (same ROIs) of layer 2/3 PFC were quantified per mouse (3 mice per group). Selected cells were manually outlined, and fluorescence of individual cells was measured using the measure stack plugin in Fiji (ImageJ 5.12h). Using the raw images, ROIs were used to outline each cell and measure parameters including area, mean fluorescence, and integrated density, along with several background fluorescence readings from adjacent regions. The corrected total cell fluorescence was then calculated as follows: Integrated Density − (Area × Mean background fluorescence).

For MBP immunofluorescence analysis, confocal tile (5 × 5) pictures of PFC layer 2/3 were taken using 40×/1.3 NA (oil immersion) objective. Four stacks were quantified per mouse. ROI of identical sizes were manually outlined to cover layer 2–3 of PFC, and fluorescence intensity was measured using the measure stack plugin in Fiji (ImageJ 5.12h).

## Reconstruction

Overview images were imported into Neurolucida 360 software (v2.8; MBF Bioscience) for axon reconstruction using the interactive tracing method with Directional Kernels. Axons were classified as myelinated if at least one MBP-positive internode was present, and as unmyelinated if no MBP-positive internodes were observed up to at least the 10th branch order. Autapses were defined as individual sites where the distance between an axon and a dendrite of the same cell was less than 1 μm. This was automatically detected using Neuroleucida after full reconstruction of the cell and identification of axons and dendrites, Autaptic connection was defined as a spatial contact between a bouton and a dendritic spine or stem. The identification of potential autaptic contacts was carried out without awareness of functional autaptic connectivity.

Following reconstruction and analysis of electrophysiological data, PV interneurons were further divided into three groups: (1) Cells with autaptic response and structural autapses, (2) cells with no autaptic response and no structural autapses, (3) cells with no autaptic response but have structural autapses. Given that the angle of sectioning during slice preparation can influence the visibility of autaptic structures, it is likely that some structural autapses were missed.

## Statistical analysis

All statistical analysis were operated using GraphPad Prism 8. First, data were tested for normality. Data sets following normal distribution were analyzed using unpaired two-tailed $t$ test. Data sets without a normal distribution were analyzed using Mann–Whitney test. One-way ANOVA with LSD test for post-hoc analysis was used for group comparison for groups with equal variances. Two-way repeated measures ANOVA were used for assessing effects within groups and between groups in experiments with repeated measurements in the same cell. All quantitative data are represented as means ± standard errors of the means (s.e.m). To reduce selection bias, all mice were randomly allocated to the different groups.

## Supporting information

**S1 Fig. Comparison between juvenile and adult demyelination. A–I**. Juvenile demyelination leads to impairment in PV interneuron maturation whereas adult demyelination induces a decrease in the excitability of PV interneurons. Summary data showing the averaged (± s.e.m) of the following intrinsic properties: (A) Resting membrane potential (RMP) (*ANOVA: n = 36,30,19,24 cells from 9/8/8/6 mice per group, \*\*\*\*p < 0.0001, post-hoc LSD test: \*p < 0.05, \*\*p < 0.01, \*\*\*\*p < 0.0001*), (B) Input resistance (IR) (*ANOVA: n = 36,30,19,24 cells from 9/8/8/6 mice per group, \*\*p = 0.0024, post-hoc LSD test: \*p < 0.05, \*\*p < 0.01, \*\*\*p < 0.001*), (C) Sag (*ANOVA: n = 36,30,19,24 cells from 9/8/8/6 mice per group, \*\*p = 0.0083, post-hoc LSD test: \*p < 0.05, \*\*p < 0.01*), (D) Rheobase (*ANOVA: n = 36,30,19,24 cells from 9/8/8/6 mice per group, p = 0.1230, post-hoc LSD test: \*p < 0.05*) and (E) Action potential (AP) threshold (*ANOVA: n = 36,30,19,24 cells from 9/8/8/6 mice per group, \*p = 0.0452, post-hoc LSD test: \*p < 0.05*), (F) AP half-width (*ANOVA: n = 36,30,19,24 cells from 9/8/8/6 mice per group, \*\*p = 0.0058, post-hoc LSD test: \*\*p < 0.01*), (G) Average action potential (AP) frequency in response to 0–650 pA current steps illustrating no significant change in S1 PV interneuron firing frequency following juvenile demyelination (*group x current two-way repeated measures: n = 36,30,19,24 cells from 9/8/8/6 mice per group: F(39,1,291) = 2.487, \*\*\*\*p < 0.0001*) and (H) Maximum firing frequency per group (*ANOVA: n = 36,30,19,24 cells from 9/8/8/6 mice per group, \*\*\*\*p < 0.0001, post-hoc LSD test: \*\*\*p < 0.001, \*\*\*\*p < 0.0001*). The data displayed in (A–F) and (H) can be found in S10 Table.
(TIFF)

**S2 Fig. Effect of AUT00201 (1 μM) on PV properties after juvenile demyelination. A-B.** Bath application of AUT00201 (1 μM) had no significant effect on the resting membrane potential (A) or the input resistance (B) of PV interneurons from cuprizone mice and control mice (RMP: *group x AUT00201 two-way repeated measures: n = 12,13 cells from 4 mice per group: p = 0.8593; AUT00201 effect: p = 0.2171*. IR: *group x AUT00201: p = 0.8605; AUT00201 effect: p = 0.4704*). **C–E.**

AUT00201 decreased the AP threshold of PV interneurons in both groups (*group x AUT00201 two-way repeated measures: n = 12,13 cells from 4 mice per group: p = 0.4503; AUT00201 effect: \*\*\*\*p < 0.0001. Post-hoc LSD analysis: \*p < 0.05, \*\*p < 0.01*) while not having an effect on either the AP amplitude (D) or the AHP (E) (D: *group x AUT00201 two-way repeated measures: n = 12,13 cells from 4 mice per group: p = 0.8235; AUT00201 effect: p = 0.0763.* E: *group x AUT00201: p = 0.6088; AUT00201 effect: p = 0.3628.* **F**. There was no effect detected on the overall firing frequency of PV interneurons from cuprizone-treated mice after bath application of AUT00201 when averaging all the cells together (*group x current two-way repeated measures: n = 13 cells from 4 mice per group: F(13,78) = 0.9280, p = 0.5287. AUT00201 effect: F(1,6) = 5.05, p = 0.0657*). The data displayed in (A–E) can be found in S10 Table.
(TIFF)

**S3 Fig. Effect of KV3 modulation on PV interneuron firing frequency and on autaptic release.** Effect of bath application of AUT00201 (1 µM) (green) on the firing frequency of individual PV interneurons in control (black) and cuprizone-treated (red) mice. The cells were divided into three groups: AUT00201 improved the firing, impaired the firing or had no effect.
(TIFF)

**S4 Fig. Effect of AUT00201 (1 µM) on PV properties after adult demyelination. A–E.** Bath application of AUT00201 (1 µM) had no significant effect on any of the intrinsic properties of PV interneurons following adult demyelination. Summary data showing the averaged (± s.e.m) of the AP threshold (A), the rheobase (B), the AP half-width (C), the AHP (D) and the overall firing frequency of PV interneurons from mice that underwent adult demyelination (E) (*Paired t test:* A: *p = 0.1006,* B: *p = 0.0678,* C: *p = 0.1006,* D: *p = 0.2812,* E: *p = 0.0653; two-way repeated measures: AUT00201 effect: F(1,12) = 0.4990, p = 0.4934*). The data displayed in (A–E) can be found in S10 Table.
(TIFF)

**S5 Fig. Autapses play a crucial role in PV interneuron's sustained firing at high frequencies. A–F.** Summary data of the averaged (± s.e.m) altered intrinsic properties of PV interneurons grouped by whether they show autaptic responses or not in both the control and the cuprizone-treated group: (**A**) Rheobase (*t test of control with vs. without autapse; n = 19 cells from 9 mice per group, \*p < 0.01; t test of cuprizone with vs. without autapse; n = 17 cells from 8 mice per group, p = 0.6246*), (**B**) Action potential (AP) threshold (*t test of control with vs. without autapse; n = 19 cells from 9 mice per group, p = 0.5680; t test of cuprizone with vs. without autapse; n = 17 cells from 8 mice per group, p = 0.7648*), (**C**) After-hyperpolarization (AHP) amplitude (*ANOVA: n = 19,19,18,18 cells from 9−8 mice per group, p = 0.0010, post-hoc LSD test: \*p < 0.05, \*\*\*\*p < 0.0001*) and (**D**) AP half-width (*ANOVA: n = 19,19,18,18 cells from 9−8 mice per group, p = 0.0002, post-hoc LSD test: #p = 0.0607 \*\*\*p < 0.001, \*\*\*\*p < 0.0001*). **E**. Average action potential (AP) frequency in response to 0−650 pA current steps illustrating a significant increase in PV interneuron firing frequency at low-current steps in control cells without autapses (gray) compared to control cells with autapses (black). Note that in cuprizone mice there was no difference (*group x current two-way repeated measures: n = 19,19,18,18 cells from 9−8 mice per group: F(39,856) = 1.671, p = 0.0068, post-hoc LSD test: \*p < 0.05*). **F**. Summary data of the maximum firing frequency per group revealing a decrease in the maximum firing frequency in PV interneurons that do not have autapses (*t test of control with vs. without autapse; n = 19 cells from 9 mice per group, \*p < 0.01; t test of cuprizone with vs. without autapse; n = 17 cells from 8 mice per group, p = 0.0722*). **G**. Pie chart showing the fraction of cells with impaired or intact firing and with or without autapses in both control mice (left) or cuprizone-treated mice (right). **H.** *Left.* Fraction of cells with intact or impaired firing within all the cells that had no autaptic response in the cuprizone group. *Right.* Fraction of cells with or without an autaptic response within all the cells that had impaired firing in the cuprizone group. The data displayed in (A–D) AND (F) can be found in S10 Table.
(TIFF)

**S1 Video. 3D reconstruction of a control PFC PV interneuron using Neurolucida showing the soma in light blue, the dendrites in green, and the axon in light blue.** The video also includes an image of the reconstructed cell filled with biocytin (orange). In this video, the autapses are marked as red circles and one of the autapses is selectively highlighted by zooming-in on it.
(MP4)

**S1 Table. All data points from Fig 1.**
(XLSX)

**S2 Table. All data points from Fig 2.**
(XLSX)

**S3 Table. All data points from Fig 3.**
(XLSX)

**S4 Table. All data points from Fig 4.**
(XLSX)

**S5 Table. All data points from Fig 5.**
(XLSX)

**S6 Table. All data points from Fig 6.**
(XLSX)

**S7 Table. All data points from Fig 7.**
(XLSX)

**S8 Table. All data points from Fig 8.**
(XLSX)

**S9 Table. All data points from Fig 9.**
(XLSX)

**S10 Table. All data points from S1 to S5 Figs as well as the supplementary method table.**
(XLSX)

## Acknowledgments

We thank Nadia Pilati, Martin Gunthorpe, Martin Pue, and Charles H. Large from Autifony S.r.l. for providing us with the Kv3 modulator AUT00201 and for their critical input on the subject; D. Slump for her assistance with breeding and genotyping of mice; D. Rotaru for her helpful suggestions; and all members of the Kushner laboratory for their support.

## Author contributions

**Conceptualization:** Sara Hijazi, Maria Pascual-García, Steven A. Kushner.

**Data curation:** Sara Hijazi, Maria Pascual-García, Yara Nabawi.

**Formal analysis:** Sara Hijazi.

**Funding acquisition:** Steven A. Kushner.

**Investigation:** Sara Hijazi, Maria Pascual-García, Yara Nabawi.

**Methodology:** Sara Hijazi.

**Project administration:** Sara Hijazi, Steven A. Kushner.

**Resources:** Steven A. Kushner.

**Supervision:** Steven A. Kushner.

**Writing – original draft:** Sara Hijazi, Maria Pascual-García, Steven A. Kushner.

**Writing – review & editing:** Sara Hijazi, Steven A. Kushner.

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
