## [Editor Report · Decision Letter 0]

29 Apr 2025

Dear Dr Kushner,

Thank you for submitting your manuscript entitled "A critical period for prefrontal cortex PV interneuron myelination and maturation" for consideration as a Research Article by PLOS Biology.

Your manuscript has now been evaluated by the PLOS Biology editorial staff, as well as by an academic editor with relevant expertise, and I am writing to let you know that we would like to send your submission out for external peer review.

Once your full submission is complete, your paper will undergo a series of checks in preparation for peer review. After your manuscript has passed the checks it will be sent out for review. To provide the metadata for your submission, please Login to Editorial Manager (https://www.editorialmanager.com/pbiology) within two working days, i.e. by May 01 2025 11:59PM.

Kind regards,

Taylor

Taylor Hart, PhD,

Associate Editor

PLOS Biology

thart@plos.org

---

## [Decision Letter · Decision Letter 1]

23 Jun 2025

Dear Dr Kushner,

Thank you for your patience while your manuscript "A critical period for prefrontal cortex PV interneuron myelination and maturation" went through peer-review at PLOS Biology. Your manuscript has now been evaluated by the PLOS Biology editors, an Academic Editor with relevant expertise, and by several independent reviewers.

In light of the reviews, which you will find at the end of this email, we are pleased to offer you the opportunity to address the comments from the reviewers in a revision that we anticipate should not take you very long. We will then assess your revised manuscript and your response to the reviewers' comments with our Academic Editor aiming to avoid further rounds of peer-review, although we might need to consult with the reviewers, depending on the nature of the revisions.

The reviewers describe the evidence presented as compelling and praise the level of interest in the findings. However, they also raised concerns over missing methodological and technical details, and limitations in several areas, including the quantification of myelination and Kv3 expression, as well as the consideration of sex specific effects. We think that the study would benefit from additional quantifications in line with the reviewers' suggestions, and would be happy to extend the revision window from one month to three months to give you time to implement this. However, if you are unable to provide these quantifications, we also think that the concerns raised could be addressed through provision of the missing elements and thorough revisions to the text.

**IMPORTANT - SUBMITTING YOUR REVISION**

*Resubmission Checklist*

*Published Peer Review*

*PLOS Data Policy*

*Blot and Gel Data Policy*

Sincerely,

Taylor

Taylor Hart, PhD,

Associate Editor

PLOS Biology

thart@plos.org

REVIEWS:

Reviewer #1: Review

The manuscript "A critical period for prefrontal cortex PV interneuron myelination and maturation" by Hijazi and colleagues addresses a highly relevant question - is the myelination of PV interneurons in the murine prefrontal cortex subject to critical periods during development? The study proceeds to very comprehensively show that yes indeed, there seems to a very relevant developmental period that shapes myelination in this cell type. To reach this conclusion, the authors employ a number of carefully conducted experiments, with the ephys recordings being the most thorough and convincing. The experiments are well-designed, statistically sound, and provide all data necessary to reach the conclusions offered in this manuscript. I find this study of significant impact, novelty, and certainly of interest for a multidisciplinary readership.

My comments mostly concern the methods section, which in my opinion could profit from more detail on experimental procedures.

1. The methods do not outline how immunofluorescence (not immunohistochemistry!) was quantified, especially in Fig. 5 where this is crucial to understand the findings regarding Kv3.1 expression changes. Using fluorescence intensity as a quantitative measure is highly problematic (a semi-quantitative method at best, and if not run with internal controls, this is hard to interpret). Please provide details for your quantification approach, including how many ROIs were analyzed.

2. The methods chapter on immunohistochemistry (which should be corrected to immunofluorescence) does not contain even the barest of protocols, and only refers to previous papers. Please outline at least the minimum information so that a reader can evaluate the procedure without having to check other publications. You do this for your ephys recordings, ("in brief, ..."), please include this here as well. Which controls were run for the antibodies?

3. The methods chapter on confocal imaging is missing important information on the objective (NA, immersion media?), and the laser lines used with this microscope. Again, please include at least a brief overview of your reconstruction procedure. If your step size for imaging z-stack was 1 µm, applying Nyquist conditions, you cannot adequately quantify structures smaller than 1 µm. Since this was apparently already done for previous publications (quantification of autapses), maybe the authors can elaborate a bit on how this important morphological detail was quantified. Maybe in the figure on autapses and their morphology, the authors could include a high res insert that shows what these putative contact points between dendrite and axon look like? It is impossible to tell from Fig. 7E as it stands now.

4. A more accurate statistical depiction would be to include STDEV instead of SEM, since you show individual cells as n, and not grouped across animals. The authors may want to reconsider this, but I leave it up to them.

Discussion

The last chapter titled "Relevance for human disorders" is a bit confusing since it is not clear right away, which of the references are work done in rodent or human. This is also something I noticed while reading the otherwise very helpful introduction (paragraph starting with "Abberant PV interneuron maturation …."). Maybe the authors could clarify which species is investigated in their references choices?

As a general suggestion - the authors may consider to switch the color scheme of all red/green figures to one that is more suitable for color vision-deficient readers. The red and green is very hard to discern, especially in those images offering overviews.

Reviewer #2: This paper studies the role of myelin on PV-neuron maturation during adolescence, a critical developmental period for PFC maturation. The results show that demyelination during the juvenile period, when the PFC and PV interneurons undergo maturation, leads to changes in PV morphology and electrophysiological properties. Specifically, cuprizone-induced demyelination is associated with compromised high-frequency firing, decrease in Kv3 currents and disrupted autaptic inhibition. These effects were only partly rescued with remyelination. Conversely, demyelination in adulthood was not associated with similar changes. Overall, this manuscript makes a significant contribution to the field and provides compelling evidence that demyelination can play a crucial role in PV-circuitry and PFC maturation, with potential implicatios to psychiatric disorders. The manusrcipt is very well-written and the conclusions of this paper are generally well-supported by the data. However, the following points require clarification.

a) The authors state that: "Juvenile cuprizone treatment led to a clear decrease in myelination in adulthood and a complete demyelination of PV interneuron axons in the PFC". Was the loss of myelin, or remyelination following cuprizone, on PV-neurons quantified somehow? Since the effects of cuprizone on demyelination can be dependent on age, strain, brain region etc, quantification of the de/remyelination would help to rule out the possibility that some of the changes are due to incomplete de/remyelination. Or if the effects of cuprizone on PV-myelination in juvenile mice are characterized elsewhere, please provide reference to it.

b) Given the extensive, and highly relevant, discussion regarding the effects of social isolation stress on PFC myelination, I assume the mice in the experiment were group housed to rule out the effects of social isolation on control and cuprizone mice. This should be clearly indicated in the materials and methods.

c) The experiments included both male and female mice. Did the authors evaluate whether there were any sex-specific effects? If sex differences were not assessed (either due to small sample sizes or other limitations), or if no differencies were present, this should be explicitly stated.

d) Kv3 immunofluorescence expression experiment is missing from the materials and methods. Was the intensity measured on the soma and/or along the axons of PV-neuron? While the electrophysiological experiments show solid evidence for altered Kv3 conduction in the model, the discussion could benefit from a short section where it is discussed how lack of myelin could lead to reduced Kv3 expression (especially if the analysis is focused on the channels on the soma).

e) The absence of selectivity for specific axons or cell types leaves open the question of whether the observed changes are driven by demyelination of PV+ axons, loss of myelin in excitatory axons, or a combination of both. This limitation should be mentioned in the discussion.

Minor issues:

References 16-18 in the introduction shown as number and not written out as the other references.

Some of part of the figures seem to be cut out, Figure 1. text of the lower panels.

And in Figure 1 experimental timeline: change regular show to regular chow.

Reviewer #3: This excellent paper breaks important new ground on the role of myelin in regulating the physiology of PV+ interneurons in neocortex. It also dents, but doesn't kill one of my favorite hypotheses on the interaction between axon length and myelination of firing patterns in these neurons in this area (all the better!). The paper shows a developmental critical period for neuronal plasticity of ion channel number/distribution, leading to significant alterations in the firing patterns of these neurons thought to be critical for a variety to animal behavior.

The study uses the well-established cuprazone model for producing significant demyelination when added to diet. It is demonstrated that demyelination during the period from P21 in mice, produces significant changes in the electrophysiological properties of PV neurons, that alter their repetitive firing patters; those patterns known to be important in behavior outcomes for the animal. Among these changes are more hyperpolarized resting potentials, decreased current sags, decreased reobase, wider action potentials, lower frequency of action potential output, particular with high strength stimulation; in particular a decrease in maximum firing frequency and less ability to sustain repetitive firing. A interesting finding is that much of this may be attributed to downregulation of KV3 potassium channels, an effect that can be partially rescued by upregulation by a positive modulator of the channel. They also demonstrate morphological alteration of PV neuron axons, with shorter axons, and most interesting, fewer autapses. These changes could be partially reversed when the neurons are allowed to remyelinate, adding additional confidence in their interpretation of the results.

Most surprising to me, was that adult demyelination produces few of the physiological changes, establishing the nature of the changes as happening within a critical developmental period. I'm surprised that there was so little change in firing frequency, given the timing alterations in action potential arrival that would be presumed to occur with even adult demyelination. But the results are clear that they isn't.

It would have been nice if they'd actually been able to measure any changes in conduction velocity with demyelination, but I'm not asking for this in the paper, because it is somewhat outside the scope of their work, and the techniques they employed were not capable of measuring it (this would require paired cell recording).

I only have minor comments, most of which I leave to the discretion of the authors to address, although I think they would improve their paper (i.e. none of these are damaging to their data, or the interpretation of same).

1. I can't find any mention of the species used, although there are clues ;). The species should be added to the title, the abstract, and species/strain to the M&M.

2. Even though the cuprazone model is well established in the field as a selective demyelinating agent, I think the interpretation of their results might be enhanced by some discussion of potential off-target effects, a worry, since the substance is administered systemically, and is known to have effect on peripheral organs, in particular the liver, which could, in turn, produce secondary metabolic effects that might produce some of the effects in brain that they describe. In this context, I'm particularly worried about the down regulation of KV3, which might simply be a correlated effect not caused by the loss of myelin. I don't see this as any kind of fatal flaw, because this is the case with many if not most pharmacological manipulations, I just think it deserves a bit more discussion

3. TEA is not all that selective for KV3, but the results are entirely consistent with the author's hypothesis. Again, maybe just some additional discussion around this issue.

4. There are a couple of places in the paper where the authors seem to imply that correlation proves causation, which of course, it doesn't. The one that sticks out, in addition to the one listed in point point 3, is the statement that loss of autapses and the changes in autapse facilitation, 'predicts' impairments in high frequency firing, when those factors only correlate with the changes in firing.

5. The most obvious explanation for the only partial recovery of physiologcal properties upon remyelination, is that recovery of myelin is also partial. I would have made a stronger argument if they had an independent means of measuring the extent of myelination, which is something that is in the capability of the approaches utilized.

6. I think that the discussion, particularly the last paragraph, would be enhanced, by inclusion of some discussion of Karl Deiserroth's papers on the effects of altering PV neuron activity on animal behavior

---

## [Editor Report · Decision Letter 2]

18 Aug 2025

Dear Dr Kushner,

Thank you for your patience while we considered your revised manuscript "A critical period for prefrontal cortex PV interneuron myelination and maturation in mice" for publication as a Research Article at PLOS Biology. This revised version of your manuscript has been evaluated by the PLOS Biology editors and the Academic Editor.

Based on our Academic Editor's assessment of your revision, we are likely to accept this manuscript for publication, provided you satisfactorily address the following data and other policy-related requests:

* We would like to suggest a different title to improve its accessibility for our broad audience:

"Transient juvenile demyelination impairs maturation and function of parvalbumin-positive interneurons in the prefrontal cortex"

* Please add the links to the funding agencies in the Financial Disclosure statement in the manuscript details.

* Please include the approval/license number of the ethical approval for the animal experiments.

* Please add a scale bar for the microscopy pictures.

* Please make sure that all figures use a colorblind-friendly palette.

* DATA POLICY:

Regardless of the method selected, please ensure that you provide the individual numerical values that underlie the summary data displayed in the following figure panels as they are essential for readers to assess your analysis and to reproduce it: 1DEGH, 2BCDEFHIJKLO, 3DEFGHJKLMNQ, 4DEFGHJKLMNQ, 5FGHI, 6AB, 7CDIJMN, 8BCD, 9BEFGHIJKLMNQVWX, S1ABCDFH, S2ABCDE, S4ABCDE and S5ABCDF.

* CODE POLICY

We expect to receive your revised manuscript within two weeks.

*Published Peer Review History*

*Press*

Sincerely,

Christian

Christian Schnell, Ph.D.

Senior Editor

PLOS Biology

cschnell@plos.org

on behalf of

Taylor Hart, PhD

Associate Editor

thart@plos.org

PLOS Biology

---

## [Editor Report · Decision Letter 3]

28 Aug 2025

Dear Dr Kushner,

Thank you for submitting your revised manuscript. However, some remaining points need to be addressed before your paper can be accepted for publication. See below for our remaining editorial requests:

----

Scale bars:

Thank you for adding scale bars to the microscopy images. However, some of the panels still lack scale bars. Please add them (eg Fig. 1C and insets; Fig. 1F left panel; Fig. 3B left panel; Fig. 4B top panel; Fig. 5E left panels; Fig. 9C top panel).

Supplemental items:

Supplementary Figure 6 and Supplementary Video 1 appear to be missing. Please provide them.

Data:

As requested in our prior decision, please provide access to the quantitative data, either in a supplementary spreadsheet or through an online repository. This applies to the data underlying the following figures, plus any similar panels that appear in Supplementary Figure 6 (as we cannot currently assess it):

1DEGH, 2BCDEFHIJKLO, 3DEFGHJKLMNQ, 4DEFGHJKLMNQ, 5FGHI, 6AB, 7CDIJMN, 8BCD, 9BEFGHIJKLMNQVWX, S1ABCDFH, S2ABCDE, S4ABCDE and S5ABCDF.

Please also add text to the figure legends indicating where the data can be found.

See below for further information about our Data Policy:

Regardless of the method selected, please ensure that you provide the individual numerical values that underlie the summary data displayed in the following figure panels as they are essential for readers to assess your analysis and to reproduce it.

Line 78: typo in the word 'Right'

----

We expect to receive your revised manuscript within two weeks.

*Published Peer Review History*

*Press*

Sincerely,

Taylor

Taylor Hart, PhD,

Associate Editor

thart@plos.org

PLOS Biology

---

## [Editor Report · Decision Letter 4]

2 Sep 2025

Dear Dr Kushner,

Thank you for implementing the additional changes that we requested. However, there still appears to be an issue with the data availability.

We see that your cover letter and new manuscript text refer to nine supplementary tables that should contain the data for each figure. However, we cannot find the actual tables/data. Can you please upload these as Supplementary Data excel sheets and ensure that these are included within your submission? Please see below for our recommendations on this point:

Supplementary files (e.g., excel). Please ensure that all data files are uploaded as 'Supporting Information' and are invariably referred to (in the manuscript, figure legends, and the Description field when uploading your files) using the following format verbatim: S1 Data, S2 Data, etc. Multiple panels of a single or even several figures can be included as multiple sheets in one excel file that is saved using exactly the following convention: S1_Data.xlsx (using an underscore).

Please reach out if you have any questions about this. We expect your revision within one week.

*Published Peer Review History*

*Press*

Sincerely,

Taylor

Taylor Hart, PhD,

Associate Editor

thart@plos.org

PLOS Biology

---

## [Editor Report · Decision Letter 5]

16 Sep 2025

Dear Dr Kushner,

Thank you for the submission of your revised Research Article "Transient juvenile demyelination impairs maturation and function of parvalbumin-positive interneurons in the prefrontal cortex" for publication in PLOS Biology. On behalf of my colleagues and the Academic Editor, Alberto Bacci, I am pleased to say that we can in principle accept your manuscript for publication, provided you address any remaining formatting and reporting issues. These will be detailed in an email you should receive within 2-3 business days from our colleagues in the journal operations team; no action is required from you until then. Please note that we will not be able to formally accept your manuscript and schedule it for publication until you have completed any requested changes.

PRESS

Sincerely, 

Taylor Hart, PhD,

Associate Editor

PLOS Biology

thart@plos.org